# Energy efficiency and biological interactions define the core microbiome of deep oligotrophic groundwater

Maliheh Mehrshad [1,2,9 ✉], Margarita Lopez-Fernandez [3,7,9 ✉], John Sundh [4], Emma Bell [5,8], Domenico Simone [3,6], Moritz Buck[2], Rizlan Bernier-Latmani [5], Stefan Bertilsson [1,2] & Mark Dopson [3]

While oligotrophic deep groundwaters host active microbes attuned to the low-end of the bioenergetics spectrum, the ecological constraints on microbial niches in these ecosystems and their consequences for microbiome convergence are unknown. Here, we provide a genome-resolved, integrated omics analysis comparing archaeal and bacterial communities in disconnected fracture fluids of the Fennoscandian Shield in Europe. Leveraging a dataset that combines metagenomes, single cell genomes, and metatranscriptomes, we show that groundwaters flowing in similar lithologies offer fixed niches that are occupied by a common core microbiome. Functional expression analysis highlights that these deep groundwater ecosystems foster diverse, yet cooperative communities adapted to this setting. We suggest that these communities stimulate cooperation by expression of functions related to ecological traits, such as aggregate or biofilm formation, while alleviating the burden on microorganisms producing compounds or functions that provide a collective benefit by facilitating reciprocal promiscuous metabolic partnerships with other members of the community. We hypothesize that an episodic lifestyle enabled by reversible bacteriostatic functions ensures the subsistence of the oligotrophic deep groundwater microbiome.

[1] Department of Ecology and Genetics, Limnology and Science for Life Laboratory, Uppsala University, Uppsala, Sweden. [2] Department of Aquatic Sciences and Assessment, Swedish University of Agricultural Sciences, Uppsala, Sweden. [3] Centre for Ecology and Evolution in Microbial Model Systems (EEMiS), Linnaeus University, Kalmar, Sweden. [4] Dept of Biochemistry and Biophysics, National Bioinformatics Infrastructure Sweden, Science for Life Laboratory, Stockholm University, Solna, Sweden. [5] Environmental Microbiology Laboratory, Environmental Engineering Institute, School of Architecture, Civil and Environmental Engineering, École Polytechnique Fédérale de Lausanne, Lausanne, Switzerland. [6] SLU Bioinformatics Infrastructure, Swedish University of Agricultural Sciences, Uppsala, Sweden. [7] Present address: Department of Microbiology, University of Granada, Granada, Spain. [8] Present address: Department of Biological Sciences, University of Calgary, Calgary, Alberta, Canada. [9] These authors contributed equally: Maliheh Mehrshad, Margarita Lopez-Fernandez. ✉email: maliheh.mehrshad@ebc.uu.se; margaritalopez@ugr.es

Archaea and Bacteria are critical components of deep groundwater ecosystems that host all domains of life as well as viruses[1–4]. With estimated total abundances of $5 \times 10^{27}$ cells[5,6], they are constrained by factors such as bedrock lithology, available electron donors and acceptors, depth, and hydrological isolation from the photosynthesis-fueled surface[6,7]. The limited number of access points to study these environments render our knowledge of deep groundwaters too patchy for robustly addressing eco-evolutionary questions. Consequently, ecological strategies and factors influencing the establishment and propagation of the deep groundwater microbiome, along with its comprehensive diversity, metabolic context, and adaptations remain elusive.

The deeply disconnected biosphere environments are subject to constant and frequent selection hurdles, which define not only the composition of the resident community, but more importantly also its strategies to cope with episodic availability of nutrients and reducing agents. In the geochemically stable and low-energy conditions characteristic for the deep biosphere, it is suggested that microbes only occasionally have access to the "basal power requirement" for cell maintenance (e.g., biomass production and synthesis of biofilms, polymeric saccharides, etc.) or the costly process of duplication[8,9]. Inspecting the expression profile and metabolic context of actively transcribing microbes may reveal the dominant ecological strategies in the deep groundwater and uncover the dimensions of its available niches. The microbial diversity of the terrestrial subsurface has previously been probed by large-scale "omics" in shallow aquifers (Rifle, USA)[10], a $CO_2$ saturated geyser (Crystal Geyser, Utah, USA)[11], and carbon-rich shales (Marcellus and Utica, USA)[12]. However, the microbiomes of extremely oligotrophic and disconnected deep groundwater ecosystems are still missing a comprehensive comparative omics analysis (further details provided in supplementary information with background knowledge on terrestrial subsurface microbiomes). The Proterozoic crystalline bedrock of the Fennoscandian Shield (~1.8 Ga. years old) hosts two sites that provide access to disconnected fracture fluids (ca. 170 to 500 meters below sea level, mbsl) running through a similar granite/granodiorite lithology[6,13–16]. The two sites, located in Sweden and Finland, provide a rare opportunity to place the microbiome of Fennoscandian Shield deep oligotrophic groundwaters under scrutiny (Supplementary Fig. S1 depicts the location of large metagenomic datasets for oligotrophic groundwaters).

In this study, we investigate the existence of a common core microbiome and possible community convergence in two extreme and spatially heterogeneous deep groundwater biomes. We leverage a large dataset that combines metagenomes, single-cell genomes, and metatranscriptomes from samples collected at two disconnected sites excavated in similar lithology in the Fennoscandian Shield bedrock. By means of an extensive genome-resolved comparative analysis of the communities, we provide support for the existence of a common core microbiome in deep groundwaters of the Fennoscandian Shield. The metabolic context and expressed functions of the microbial community were further used to elucidate the ecological and evolutionary processes essential for successfully occupying and propagating in the available niches of these extreme ecosystems.

## Results and discussion

**Fennoscandian shield genomic database (FSGD).** The Fennoscandian Shield bedrock contains an abundance of fracture zones with different groundwater characteristics that vary in water source, retention time, chemistry, and connectivity to surface-fed organic compounds (see Supplementary Data 1). The Äspö Hard Rock Laboratory (HRL) and Olkiluoto drillholes were sampled over time, covering a diversity of aquifers representing waters of differing ages and both planktonic and biofilm-associated communities. In order to provide a genome-resolved view of the Fennoscandian Shield bedrock Archaeal and bacterial communities, collected samples were used for an integrated analysis by combining metagenomes ($n = 44$), single-cell genomes ($n = 564$), and metatranscriptomes ($n = 9$) (see detailed statistics for the generated datasets in the Supplementary Data 1 and Supplementary Information). Assembly and binning of the 44 metagenomes (~1.3 TB sequenced data) resulted in the reconstruction of 1278 metagenome-assembled genomes (MAGs; ≥50% completeness and ≤5% contamination). By augmenting this dataset with 564 sequenced single-cell amplified genomes (SAGs; 114 of which were ≥50% complete with ≤5% contamination), we present a comprehensive genomic database for the archaeal and bacterial diversity of these oligotrophic deep groundwaters, hereafter referred to as the Fennoscandian Shield genomic database (FSGD; statistics in Fig. 1A & Supplementary Data 2). Phylogenomic reconstruction using reference genomes in the Genome Taxonomy Database (GTDB-TK; release 86) shows that the FSGD MAGs/SAGs span most branches on the prokaryotic tree of life (Fig. 2). Harboring representatives from 53 phyla (152 archaeal MAGs/SAGs in 7 phyla and 1240 bacterial MAGs/SAGs in 46 phyla), the FSGD highlights the remarkable diversity of these oligotrophic deep groundwaters. Apart from the exceptional case of a single-species ecosystem composed of 'Candidatus Desulforudis audaxviator' in the fracture fluids of an African gold mine[17], other studies of deep groundwaters as well as aquifer sediments have also revealed a notable phylogenetic diversity of the Archaea and Bacteria[10,11,18]. For example, metagenomic and single-cell genomic analysis of the $CO_2$-driven Crystal geyser (Colorado Plateau, Utah, USA) resulted in reconstructed genomes of 503 archaeal and bacterial species distributed across 104 different phylum-level lineages[11].

Clustering reconstructed FSGD MAGs/SAGs into operationally defined prokaryotic species (≥95% average nucleotide identity (ANI) and ≥70% coverage) produced 598 genome clusters. Based on the GTDB-TK affiliated taxonomy, a single FSGD cluster may represent a novel phylum, whereas at the lower taxonomic levels, the FSGD harbors genome clusters representing seven novel taxa at class, 58 at order, 123 at family, and 345 at the genus levels. In addition, more than 94% of the reconstructed MAGs/SAGs clusters ($n = 568$) represent novel species with no existing representative in public databases (Supplementary Data 2). Mapping metagenomic reads against genome clusters represented exclusively by SAGs ($n = 38$, Fig. 1A) revealed that 14 genome clusters (20 SAGs) were not detectable in the metagenomes, suggesting they might represent rare species in the microbial community of the investigated deep groundwaters (Supplementary Data 3).

To explore the community composition of different groundwaters and their temporal dynamics, presence/absence patterns were computed by competitively mapping the metagenomics reads against all reconstructed MAGs/SAGs of the FSGD. Contigs were discarded from the mapping results if < 50% of base pairs were covered by mapped reads. The mapping rates of the present contigs were then normalized for sequencing depth in each metagenome as TPM (transcripts per kilobase million). Since metagenomes were in some cases amplified because of low DNA amounts, we only discuss binary presence/absence when referring to the community composition to avoid inherent biases in abundance values calculated by counting mapped metagenomic reads. The Äspö HRL metagenomics samples were collected over three years from 2013–2016 from five different boreholes. The Olkiluoto metagenomics samples were collected between June and November 2016 from three different drillholes. Communities

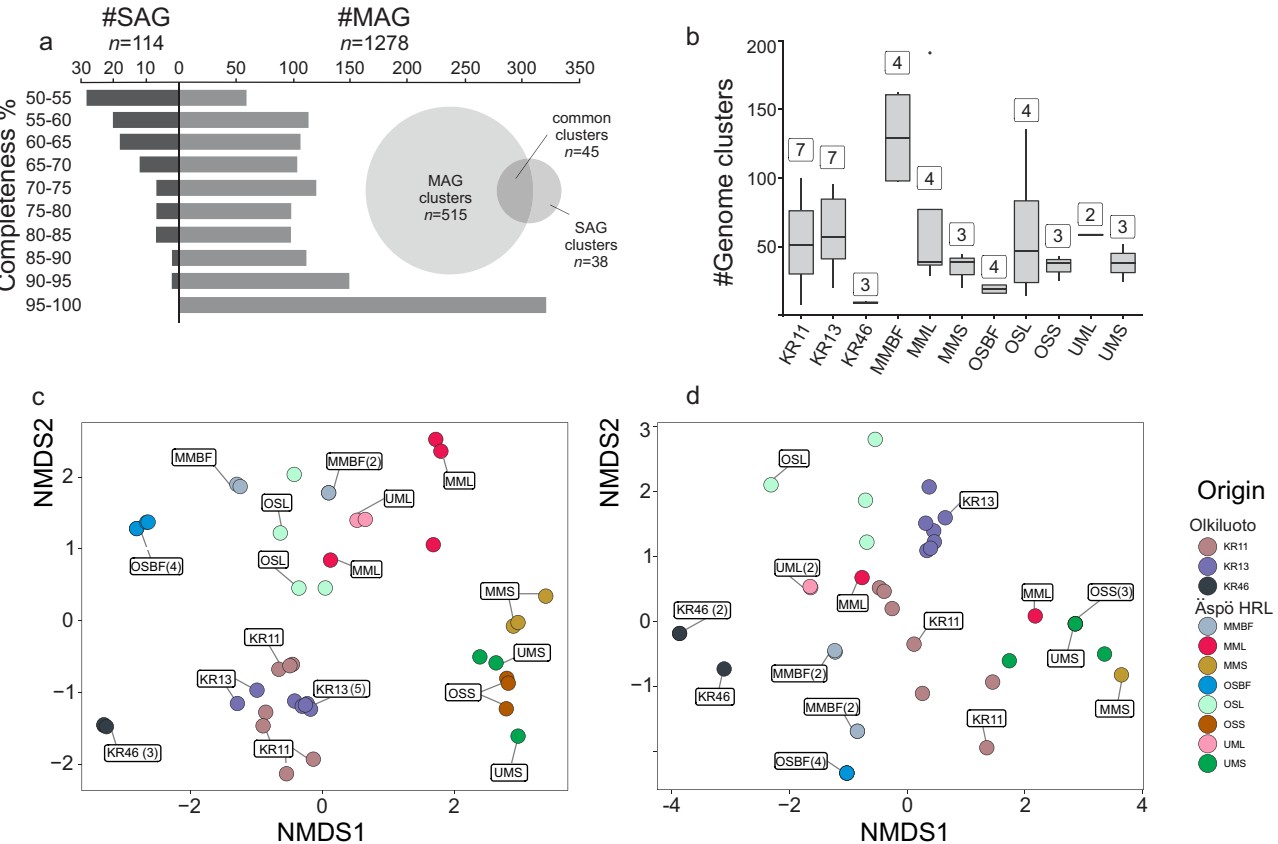

**Fig. 1 Overview of the FSGD MAGs and SAGs.** Statistics of the metagenome-assembled genomes (MAGs) and single-cell amplified genomes (SAGs) of the Fennoscandian Shield Genomic Database (**a**). The number of genome clusters present in borehole samples (centerline, median; hinge limits, 25 and 75% quartiles; whiskers, 1.5x interquartile range; points, outliers). Numbers on top of each box plot represent the number of metagenomes generated for borehole samples (**b**). NMDS plot of unweighted binary Jaccard beta-diversities of presence/absence of all FSGD reconstructed MAGs/SAGs (**c**) and MAG and SAG clusters belonging to the common core microbiome present in both Äspö HRL and Olkiluoto (**d**). Numbers in the parenthesis show the number of overlapping points. The data used to generate these plots are available in Supplementary Data 4 and the Source Data.

from each separate borehole cluster together and show only minimal variation in prokaryotic composition over time, hinting at the high stability of prokaryotic community composition in the groundwater of the different aquifers. In contrast, the different boreholes feature discrete community compositions (Fig. 3 & 1C). The observed compositional differences are likely to be at least partially caused by the varying availability of reducing agents and organic carbon in different boreholes (physicochemical characteristics are provided in Supplementary Data 1), resulting from contrasting retention time, depth, and isolation from surface inputs of organic compounds[16,19]. In the case of Äspö HRL datasets, different samples (planktonic vs. biofilm-associated microbes) or size fractions (large vs small cell size fraction) of samples originating from the same boreholes also cluster separately (Fig. 1C, sample description in the Supplementary Information document).

**Common core microbiome of the deep groundwater.** Mapping the metagenomic reads against the FSGD and removing contigs with < 50% of the base pairs covered by reads identified 165 MAGs/SAGs that were present in groundwater samples at both sites (Fig. 3 & 1D). These prevalent MAGs/SAGs, group into 73 genome clusters taxonomically affiliated to both domain Archaea (phyla Nanoarchaeota (class Woesearchaeia) and Thermoplasmatota; $n = 15$) and Bacteria (phyla Acidobacteriota, Actinobacteriota, Bacteroidota, Caldatribacteriota, Campylobacterota,

CG03, Chloroflexota, Desulfobacterota, Firmicutes_A, Nitrospirota, Omnitrophota, Patescibacteria, and Proteobacteria; $n = 150$). See Supplementary Fig. S2 and Supplementary Data 4 for details of the common core microbiome. These presence/absence patterns of the core microbiome were also supported by the assembly of representatives of the same genome cluster from both sites (15 clusters) (Supplementary Data 4).

While the disconnected nature of the two sites studied here is reflected in their discrete prokaryotic community composition (Fig. 3 & 1C), the two locations harbor bedrock with similar granite/granodiorite lithologies[6] (see Supplementary Information for a detailed description of site lithologies). Consequently, they are likely to provide similar niches that may result in convergent species incidence. The shared presence of species in both of these disconnected deep groundwaters, where the bedrock lithology is not the pressing divergence force, supports the existence of a deep groundwater core microbiome primed to occupy the fixed niches available in these two ecosystems. Further exploration of the publicly available genomes/MAGs revealed lineages where all their available genomic representatives originate exclusively from globally distributed groundwater samples, including phylum UBA9089 (Supplementary Data 5). We recovered 14 MAGs and 1 SAG belonging to this taxon in our FSGD dataset and found it to be one of the highly transcribing taxa in the metatranscriptomes. This further strengthens the concept of a deep biosphere core microbiome that reaches beyond systems with similar lithologies. However, the presence of this core

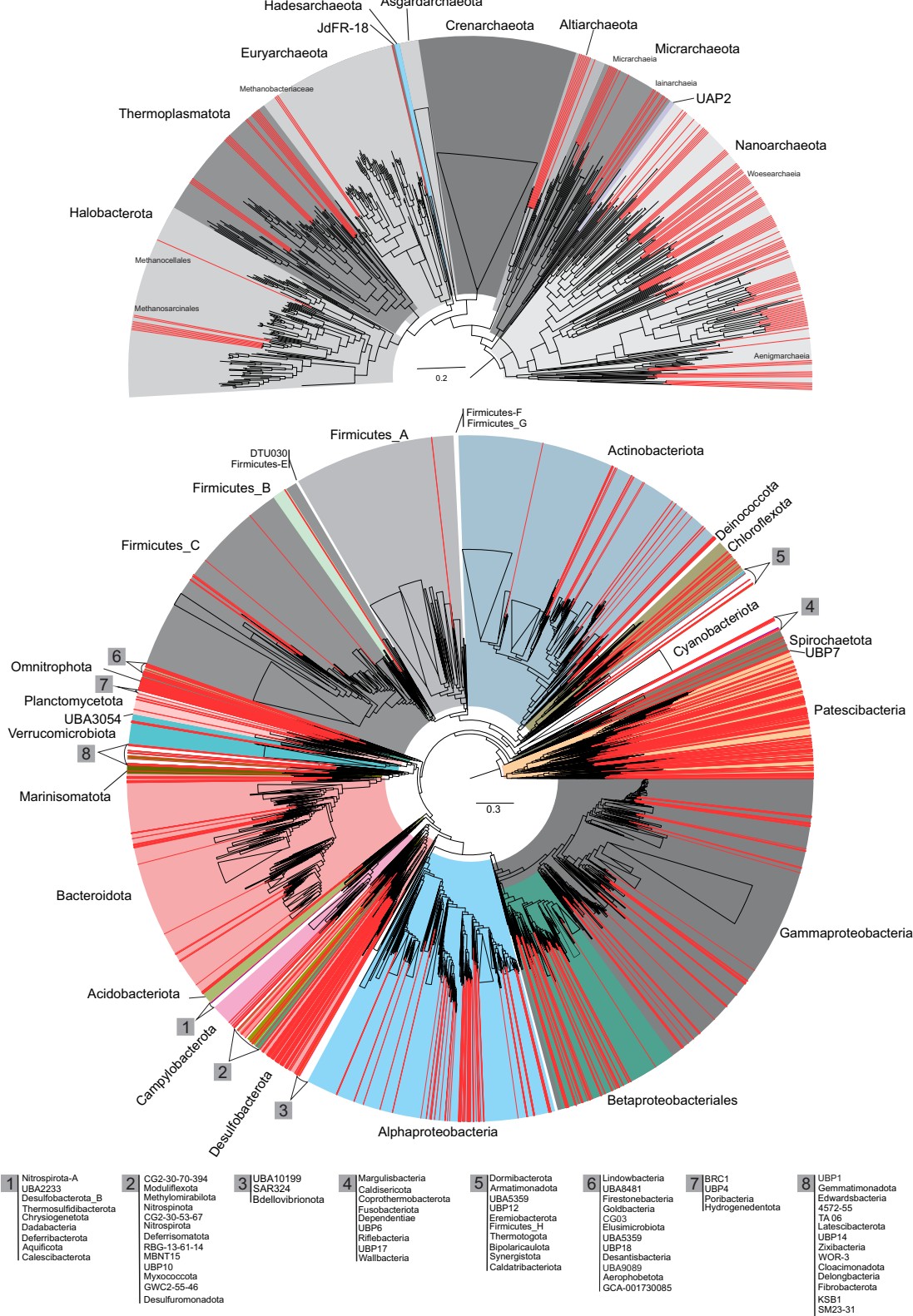

**Fig. 2 Phylogenetic diversity of reconstructed MAGs and SAGs of the fennoscandian shield genomic database (FSGD).** Genomes present in genome taxonomy database (GTDB) release 86 were used as reference. Archaea and Bacteria phylogenies are represented separately in the top and bottom panels, respectively. MAGs and SAGs of the FSGD are highlighted in red. Legend in front of each number at the bottom of the figure shows the list of taxa in the tree that are marked with the same number.

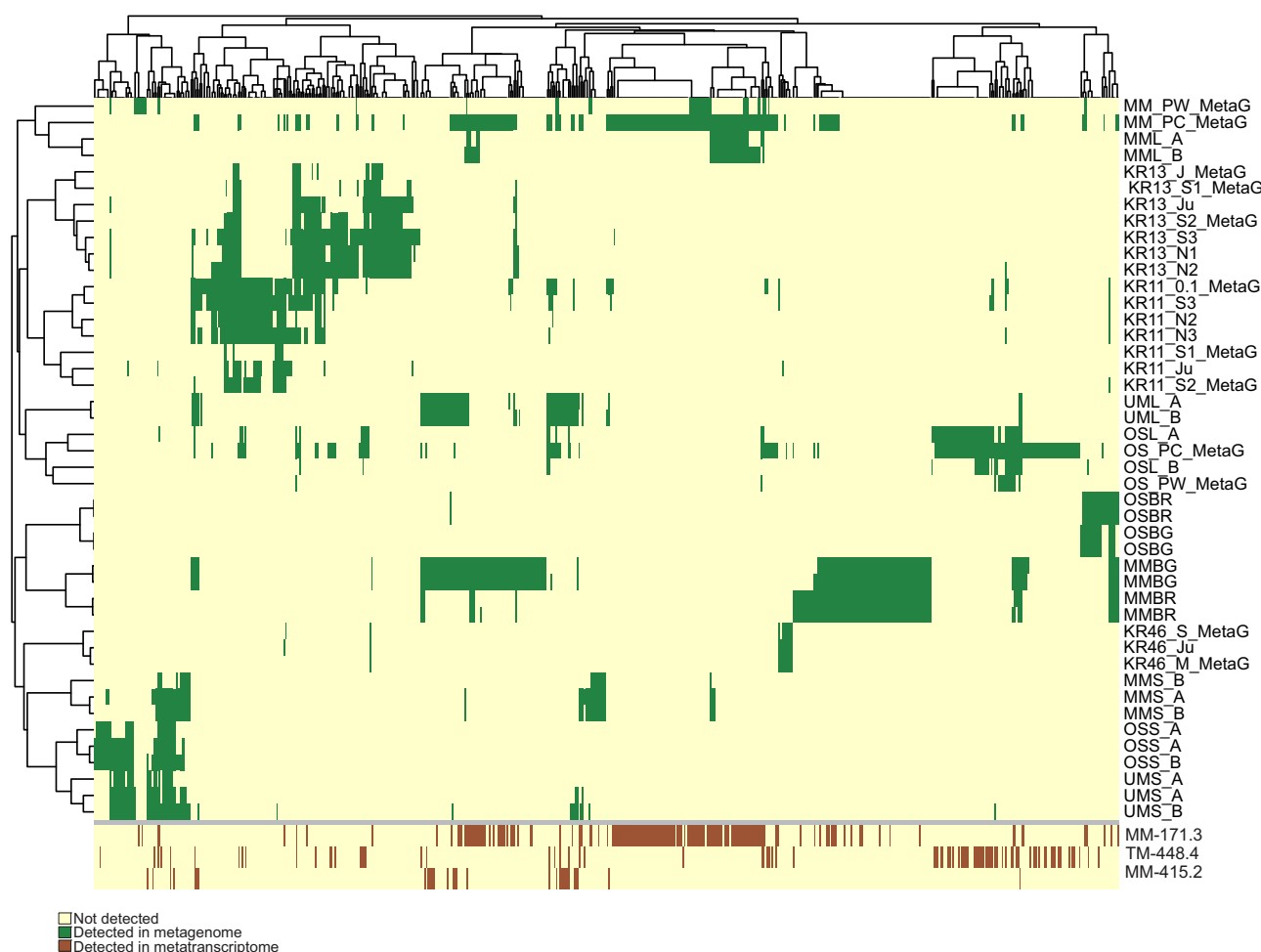

**Fig. 3 Distribution pattern and transcription status of the FSGD genome clusters along different metagenomes and metatranscriptomes.** Each column represents a genome cluster of reconstructed MAGs/SAGs of the deep groundwater datasets of this study. The top heatmap depicts the presence (colored in green) of each genome cluster along metagenomics datasets originating from different boreholes of the two sampled sites. The bottom heat map represents the transcription status of the genome cluster representatives in the metatranscriptomes originating from different boreholes of the Äspö HRL (brown). Clusters with all zero values have been removed from the plot (in total 14 clusters that are solely represented by SAGs).

microbiome in different deep groundwater ecosystems would vary due to physicochemical[20] and geological factors that affect the composition of the local biomes.

The relatively high phylogenetic diversity of the common core microbiome of the two deep Fennoscandian Shield groundwaters under scrutiny (165 genome clusters classified in 15 different phyla, 25 classes, and 41 orders) implies a significant role of ecological convergence (due to e.g., availability of nutrients and reducing agents) rather than evolutionary responses. However, we cannot yet confute the possibility of an evolutionary convergence as the community clearly undergoes adaptation over the long residence time's characteristic for deep groundwaters. For instance, salinity is a proxy for water retention time, and ranges from 0.4 (similar to the brackish Baltic proper) to 1.8% (ca. half that of marine systems) in our sampled groundwaters (Supplementary Data 1). This is reflected in a shift in the isoelectric point of predicted proteomes. The decreased prevalence of basic proteins could be a potential adaptation strategy in active groundwater microbes to their surrounding matrix over their long residence time (Fig. 4). A Kolmogorov–Smirnov test shows significantly different distributions of isoelectric points for all samples ($p < 10^{-26}$, for all comparisons). The isoelectric point trend towards a reduced prevalence of basic proteins with increasing salinity[21] is specifically pronounced in the Olkiluoto

drillhole OL-KR46 where salinity reaches a maximum of 1.8% (Fig. 4B). The relative frequency of calculated isoelectric points for predicted proteins in metagenomes sequenced from other drillholes in Olkiluoto and Äspö HRL (ranging from 0.4–1.2% salinity) showed a higher signal of basic proteins compared to OL-KR46. The pairwise Kolmogrov's $D$ values suggest a higher distribution similarity between OL-KR11 and OL-KR13 compared to OL-KR46 as well as a higher distribution similarity between MM, OS, and UM (see Supplementary Data 6 for pairwise values). The OL-KR46 drillhole provides access to fracture fluids at ~530.6 mbsl where reconstructed genome clusters have relatively low phylogenetic diversity (9–10 genome clusters per dataset and 11 unique clusters in total), thereby representing a community composition that is distinct to those from other groundwaters (Fig. 1B, C, & D). The diversity of the metagenomes was also assessed using the *gyrB* gene diversity (Supplementary Fig. S3 and Supplementary information for methods), further highlighting the low species richness of OL-KR46 compared to all other collected samples.

Inspecting the metabolic context of the reconstructed MAGs/SAGs from OL-KR46 suggests a flow of carbon between sulfate-reducing bacteria as the predominant metabolic guild in the community and acetogens, methanogens, and fermenters in agreement with earlier reported results[22]. Despite the unique

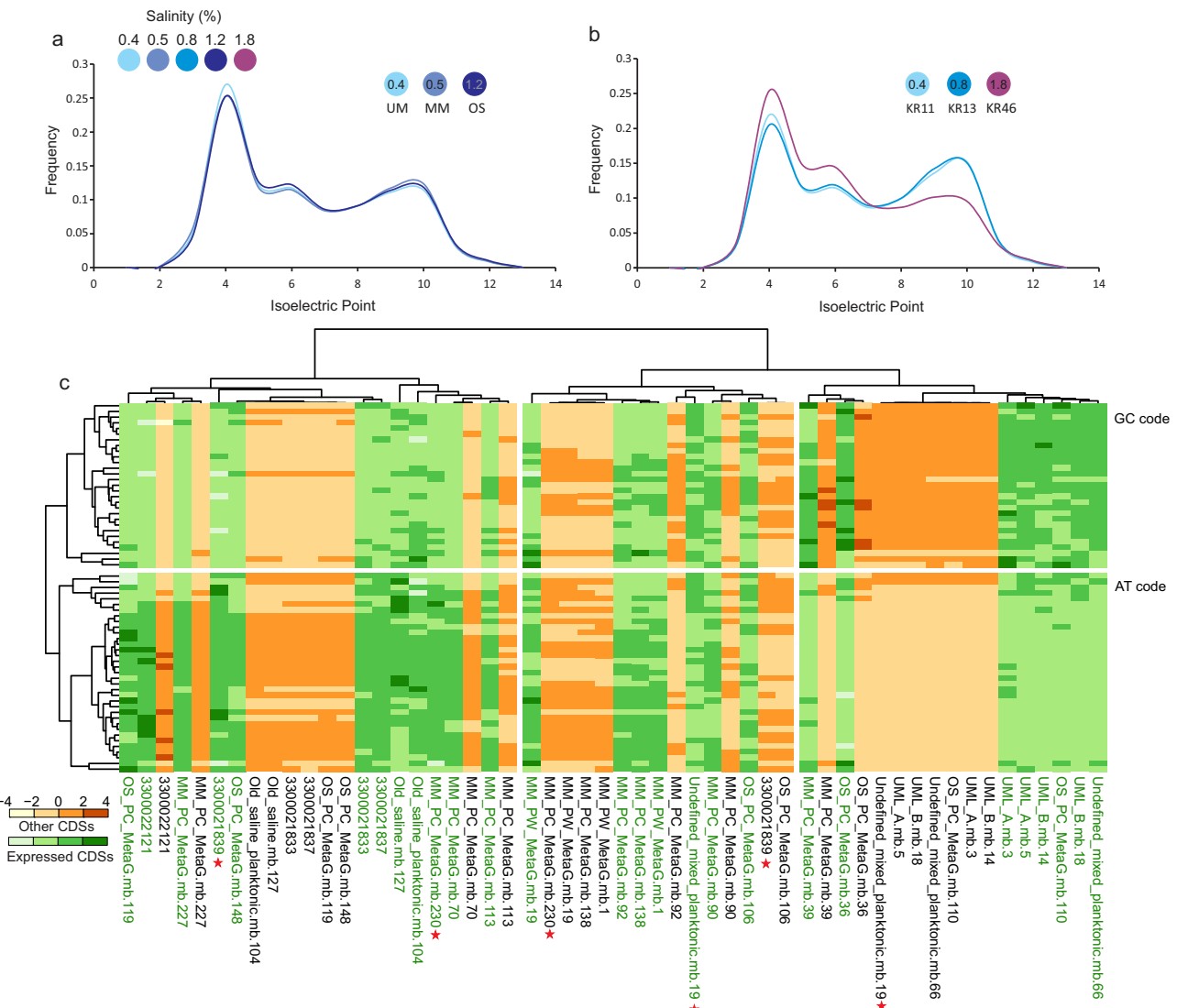

**Fig. 4 Adaptations of the coding sequences of the deep groundwater microbiome.** Relative frequency of isoelectric points in the predicted proteins of assembled metagenomes from Äspö HRL boreholes (**a**) and Olkiluoto (**b**). Data used to generate these plots are available in the Source Data. The salinity of the water flowing in each borehole is shown on the top left legend. Representation of frequency (the expected number of codons, given the input sequences, per 1000 bases) of the utilization of synonymous codons across MAGs and SAGs of different genomes clusters of highly expressing Desulfobacterota. Codon usage frequencies are calculated separately for the CDSs for which RNA transcripts are detected (green) and the rest of CDSs not actively being transcribed in the sequenced metatranscriptomes (orange) (**c**). Stars showcase potential transcription efficiency control via codon usage bias.

community composition in this aquifer, 27% of the genome clusters represented in the OL-KR46 drillhole are part of the common core microbiome present in groundwaters collected from both sites (Supplementary Data 4). Additionally, genome Cluster 25 includes seven representatives of *Pseudodesulfovibrio aespoeensis* (originally classified as *Desulfovibrio aespoeensis*)[23] that was originally isolated from 600 mbsl in Äspö HRL groundwater. Genome cluster 25 affiliated to this species contains MAGS from the Olkiluoto drillholes OL-KR13 and OL-KR46 extending the presence of representatives of this species to both studied sites.

**Metabolic potential and biological interactions.** The metabolic context and ecosystem functioning of the resident prokaryotes can provide clues about the features of the fixed deep groundwater niches and whether they are predominantly defined by biotic interactions or abiotic forces. Prior studies have proven that

representatives of all domains of life are actively transcribing in the deep groundwater ecosystems[1,22,24,25]. However, a comprehensive taxonomic and metabolic milieu of the transcribing constituents of the active Fennoscandian Shield community has not been explored. A high-resolution and genome-resolved view of the transcription pattern by mapping metatranscriptomic reads against the FSGD MAGs/SAGs was generated in this study, where actively transcribing genomes, their transcribed genes, and overall metabolic capability was catalogued. This analysis revealed that a small yet significant portion of FSGD MAG/SAG clusters is actively transcribing in the nine sequenced metatranscriptomes derived from the three Äspö HRL boreholes (ca. 26% of genome clusters in MM-171.3, 14% in TM-448.4, and 5% in MM-415.2 metatranscriptomes; Fig. 3). Resources and energy costs dedicated to protein synthesis (i.e., transcription and translation) appear sufficiently large for prokaryotes to be recognized by natural selection and accordingly impact the fitness of these prokaryotes at large[26,27]. Consequently, evolutionary adaptations such as

codon usage bias and regulatory processes are in place to adjust cellular investments in protein synthesis. Unequal utilization of synonymous codons can have implications for a range of cellular and interactive processes, such as mRNA degradation, translation, and protein folding[28,29], as well as viral resistance mechanisms and horizontal gene transfer[30,31]. Calculating the frequency of synonymous codons in the FSGD MAGs/SAGs (Supplementary Fig. S4) and those belonging to the highly expressing genome clusters (TPM > 10,000 arbitrary thresholds; Supplementary Fig. S5) revealed variable utilization of synonymous codons in different MAGs/SAGs. These variable patterns are primarily related to the range of GC content (Supplementary Figs. S4 & S5). Further exploration of the variable codon utilization among highly expressing representatives of MAGs/SAGs affiliated with phylum Desulfobacterota by separately calculating codon frequency for expressing CDSs (according to the mapping results of metatranscriptomes) highlights cases of potential transcription efficiency control via codon usage bias in their genomes (Fig. 4). While in most cases, both expressed CDSs and the rest of the CDSs in the genome represent similar synonymous codon frequency and distribution, some genomes display notable differences (e.g. MAGs undefined_mixed_planktonic.mb.19 and MM_PC_MetaG.mb.230 along with SAG 3300021839). The expressed CDSs of these reconstructed genomes encode different functions related to their central role in dissimilatory sulfur metabolism and regulatory functions (Supplementary Data 7). The expressed CDSs of the SAG 3300021839 code for heritable host defense functions against phages and foreign DNA (i.e., CRISPR/Cas system-associated proteins Cas8a1 and Cas7). Naturally, these processes are active in the case of exposure to viruses or other mobile genetic elements. Accordingly, the observed differential codon usage frequency of expressed genes compared to other genes in this SAG could hint at a potential role of codon usage bias in regulating the efficiency of translation. The same SAG also expressed the ribonuclease toxin, BrnT of type II toxin-antitoxin system. This toxin is known to respond to environmental stressors and cease bacterial growth by rapid attenuation of protein synthesis most likely via its ribonuclease activity[32]. The expression profile of this SAG serves as an example of the importance of selective transcription in this deep groundwater microbe to fine-tune its response to selfish genetic elements.

In deep groundwater ecosystems, organisms respond to limited energy and nutrient availability by adjusting their energy investment in different expressed traits (see above). Highly expressing cells are presumed to be either equipped with efficient metabolic properties and biotic interactions that are tuned to available niches (including but not limited to the dimensions of fixed niches available to the common core microbiome) or alternatively represent an ephemeral bloom profiting from sporadically available nutrients. To shed light on this, we explored the metabolic context and lifestyle of 86 FSGD genome clusters with high transcription levels (TPM ≥ 10,000, arbitrary threshold) comprising 192 MAGs and 35 SAGs. The relatively large phylogenetic diversity of these highly expressing clusters distributed across 17 phyla reaffirms that a considerable fraction of the deep groundwater microbiome has competitive properties with regards to both metabolism and interactions (Fig. 5).

Microbial public goods are loosely defined as functions or products that are costly to produce for an individual and provide collective benefit for the surrounding community. While these common goods can have different forms, they are generally released into the extracellular environment[33]. The normalized count of mapped metatranscriptomic reads on genes annotated with functions related to the provision of public goods comprises

a considerable proportion of the overall transcription profile (ca. 2–6% with one case reaching as high as 23%) in deep groundwater metatranscriptomes (see Fig. 6 & Supplementary Data 8 for list of explored K0 identifiers). One way to alleviate the "tragedy of the commons" imposed by the production and sharing of such public goods is by the emergence of local sub-communities through biofilm or aggregate formation[34]. Functions related to biofilm formation are detected in the transcription profiles (0.5–2% of the total transcription profile) of the deep groundwaters studied here. Biofilm formation could potentially help with reducing the number of microbes that profit from common goods without contributing to these shared resources and provides an evolutionary advantage for cooperation as compared to competition[34]. However, exploitation of public goods seems inevitable in groundwaters considering the high phylogenetic diversity and widespread presence of e.g. Patescibacteriota representatives (300 MAGs and 19 SAGs forming 152 clusters) as well as DPANN representatives[35] including Nanoarchaeota representatives (100 MAGs in 56 clusters in classes Aenigmarchaeia, Nanohaloarchaea, CG03, and Woesearchaeia), and Micrarchaeota (15 MAGs in 9 clusters). The metabolic context of these reconstructed MAGs/SAGs suggests a primarily heterotrophic and fermentative lifestyle lacking core biosynthetic pathways for nucleotides, amino acids, and lipids. This is in agreement with prior reports on Patescibacteriota and DPANN representatives[36] that are highly dependent on their adjacent cells to supply them with metabolites they cannot synthesize themselves[36–38]. Members of Altiarchaeota among the DPANN have been shown to be autotrophs capable of fixing carbon dioxide by using a modified version of the reductive acetyl-CoA (Wood-Ljungdahl) pathway[39]. However, FSGD MAGs and SAGs affiliated to the Altiarchaeota phylum (6 MAGs and 2 SAGs in 2 clusters) are suggested to be heterotrophic as they lack the full Wood-Ljungdahl pathway. This could be due to MAG/SAG incompleteness or that this metabolism is not consistently present in all representatives of this phylum.

The high RNA transcript counts for representatives of these phyla in the Fennoscandian Shield datasets suggest that they have sufficient energy at their disposal to carry out transcription (16 Patescibacteriota and 3 Nanoarchaeota (Woesearchaeia and Aenigmarchaeia) clusters among highly transcribing clusters; Fig. 5). Captured transcripts of representatives from these phyla were annotated as ribosomal proteins, cell division proteins (FtsZ), DNA polymerase, DNA gyrase, ATP synthase subunit alpha, glyceraldehyde-3-phosphate dehydrogenase, fimbrial protein PulG superfamily, D-lactate dehydrogenase, ribonuclease Y, elongation factor Tu, and other hypothetical proteins according to conserved domain inspections. Hence, to stabilize their own symbiotic niches, these cells likely participate in reciprocal partnerships where they supply fermentation products (e.g., lactate, acetate, and hydrogen), vitamins, amino acids, and secondary metabolites to their direct or indirect partners in their immediate surroundings. Representatives of these phyla are also detected in the common core microbiome across both groundwater sites (14 Nanoarchaeota; class Woesearchaeia MAGs and 48 Patescibacteriota MAGs/SAGs; Supplementary Data 4). This implies a significant role of symbiotic interactions in the development of fixed niches in the deep groundwaters. The epi-symbiotic association of Patescibacteriota and DPANN Archaea with prokaryotic hosts has already been verified for several representatives[20,40–44]. However, the level and range of host/partner specificity for these associations remain understudied. The incidence of the same genome clusters of Patescibacteriota and Nanoarchaeota representatives in both deep groundwaters, combined with their high expression potential and inferred

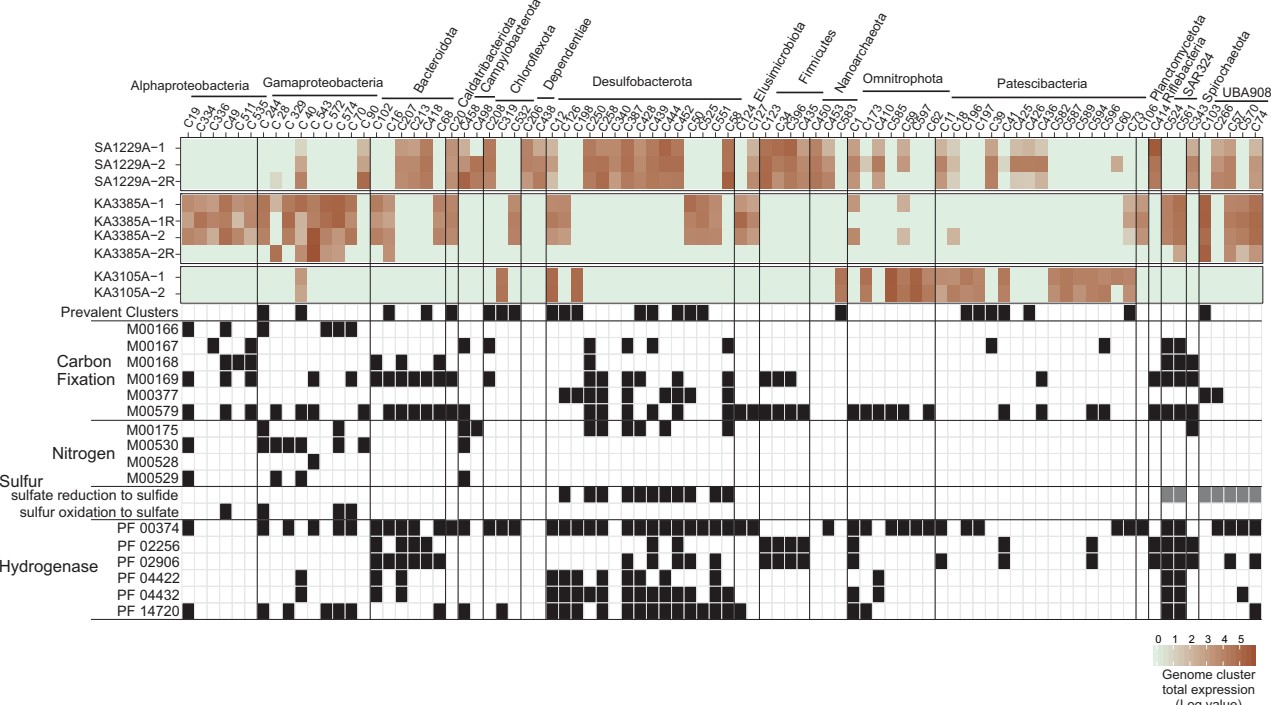

**Fig. 5 Expression profile and metabolic context of the highly expressing genomic clusters.** The expression profile of genome clusters with a total expression ≥ 10,000 TPM and their metabolic potential for nitrogen and sulfur energy metabolism as well as carbon fixation is represented as KEGG modules presence only if all genes of the module or the key genes of the process are present (in black). Reductive pentose phosphate cycle, ribulose-5P ≥ glyceraldehyde-3P (M00166), reductive pentose phosphate cycle, glyceraldehyde-3P ≥ ribulose-5P (M00167), Crassulacean acid metabolism, dark (M00168) and light (M00169), reductive acetyl-CoA pathway (Wood-Ljungdahl pathway) (M00377), and phosphate acetyltransferase-acetate kinase pathway, acetyl-CoA ≥ acetate (M00579). Nitrogen fixation (M00175), dissimilatory nitrate reduction (M00530), nitrification (M00528), and denitrification (M00529). Genes involved in the dissimilatory sulfur metabolism are shown in Supplementary Data 7. Squares highlighted in gray contain genes for sulfate reduction to sulfide but are missing dsrD. Nickel-dependent hydrogenase (PF00374), iron hydrogenase (PF02256 and PF02906), coenzyme F420 hydrogenase/dehydrogenase (PF04422 and PF04432), and NiFe/NiFeSe hydrogenase (PF14720).

dependency on symbiotic associations with other microbes for survival[38], underscores cooperation as a competent evolutionary strategy in oligotrophic deep groundwaters.

Reconstructing the metabolic scheme of highly expressing genome clusters, apart from small heterotrophic cells with a proposed symbiotic lifestyle (Fig. 5), highlights a central role of sulfur as an electron acceptor that is commonly used in the energy metabolism of the deep groundwater microbiome[25]. A total of 189 MAGs/SAGs contain the dissimilatory sulfite reductase A/B subunits (dsrAB), of which 48 branches together with the oxidative and 141 clusters with the reductive dsrA reference proteins in the reconstructed phylogeny (Supplementary Fig. S6). We further inspected these MAGs and SAGs for genes related to sulfur metabolism (sulfate adenylyltransferase (sat), adenylylsulfate reductase subunits A/B (aprAB), dissimilatory sulfite reductase A/B subunits (dsrAB), dsrC, dsrD, and dsrEFH) to determine the direction of dissimilatory sulfur metabolism in these organisms[45]. Oxidative dsrA genes belonged to 19 genome clusters affiliated to the Proteobacteria phylum. Among these, 17 clusters contribute to the sulfur cycle via sulfur oxidation to sulfate, and genome cluster 85 represents the genomic capacity for sulfur oxidation to sulfite. Genome Cluster 329 is missing dsrEFH genes and consequently, the direction of its sulfur metabolism could not be confidently assigned from the genomic content of reconstructed SAG (Supplementary Data 9).

Among MAGs and SAGs containing the reductive type of dsrA, 115 MAGs/SAGs affiliated to phyla Chloroflexota, Desulfobacterota, Firmicutes-B, and Nitrospirota contribute to the sulfur cycle via sulfate reduction to sulfide (Supplementary

Data 9) whereas for five Desulfobacterota representatives the genomic information could not differentiate between sulfite reduction to sulfide and sulfur disproportionation. A total of 21 MAGs/SAGs (affiliated to phyla AABM5-125-24, Actinobacteriota, Desulfobacterota, Nitrospirota, SAR324, UBA9089, UBP1, and Zixibacteria) contain the genes aprAB, sat, and the reductive type of dsrA but are missing dsrD. While the absence of dsrD gene in these MAG/SAG could be due to incompleteness, representatives of the listed taxa have previously been reported to be capable of sulfite/sulfate reduction[45] and feature DsrA proteins that branch together with the reductive DsrA reference proteins in our reconstructed phylogeny (Supplementary Fig. S6 and Supplementary Data 9). Representatives of 18 clusters (ca. 21%) of the highly expressing groups represent the genomic capacity for dissimilatory sulfur metabolism (Supplementary Data 9). Their complete RNA transcript profile apart from functions related to dissimilatory sulfur metabolism (aprAB, dsrAB) are related to a wide range of cellular functions. These include genetic information processing, central carbohydrate turnover, lipid and protein metabolism, biofilm formation, membrane transporters, and other cell maintenance functions as well as genes involved in replication and repair of the genome and cell division. Sulfate-reducing bacteria of the phyla Desulfobacterota and UBA9089 also contain genes encoding for the reductive acetyl-CoA pathway (Wood-Ljungdahl pathway) that is utilized in reverse to compensate for the energy-consuming oxidation of acetate to $H_2$ and $CO_2$ with energy derived from sulfate reduction[46]. In this process, cells are able to use acetate as a carbon and electron source (Fig. 5) with various hydrogenase types offering molecular

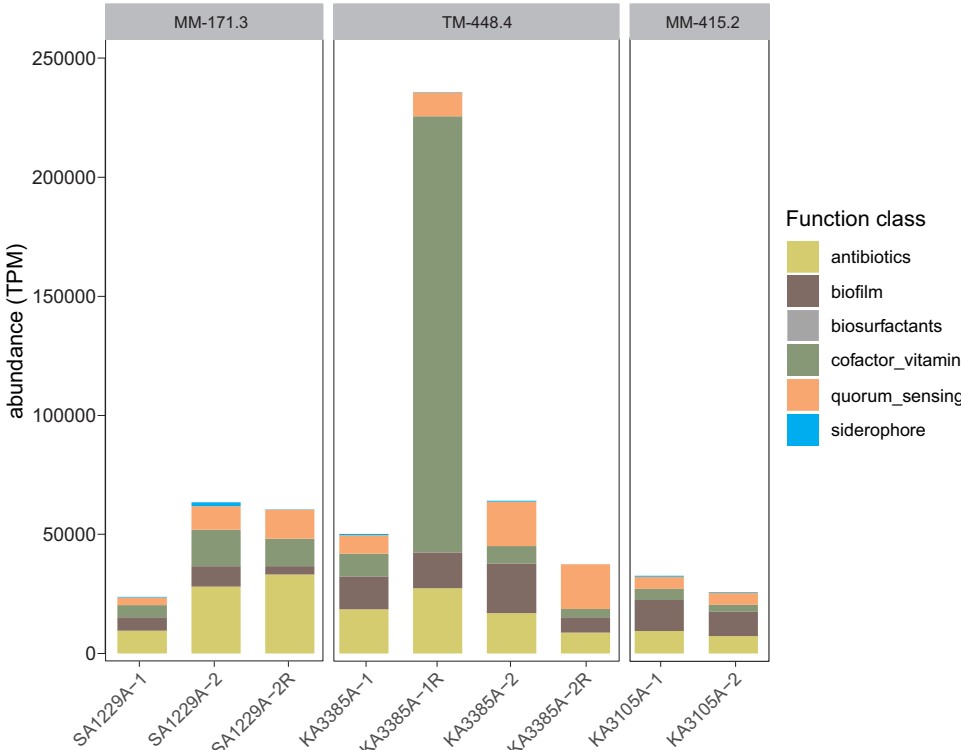

**Fig. 6 Expression level of some functional classes involved in public good provision in the sequenced metatranscriptomes.** The list of screened KO identifiers is shown in Supplementary Data 8.

hydrogen as an alternative electron donor[47]. Representatives of four highly expressing Patescibacteriota genome clusters (Clusters 41, 436, 587, and 596) contain genes encoding for phosphate acetyltransferase and acetate kinase functions. These proteins facilitate the production of acetate from acetylCoA via a two-step pathway where acetyl phosphate occurs as an intermediate. A fifth highly expressing Patescibacteriota (cluster 594) may also feature this pathway but only the acetate kinase gene was detected (annotations provided in Supplementary Data 10). The production of acetate by Patescibacteriota genome clusters could potentially supply Desulfobacterota and UBA9089 representatives with a primary carbon and electron source and form the basis for a reciprocal partnership.

A notable portion of FSGD transcripts (1.6–10% in different metatranscriptomes) belong to motility-related genes (e.g., chemotaxis and flagella assembly) and studies show that the expression of this costly trait increases in low nutrient environments as an adaptation to anticipate and exploit nutrient gradients[48]. In addition to motility-related genes, sulfate-reducing representatives of the FSGD invest in the transcription of type IV secretion systems that can facilitate adhesion, biofilm formation, and protein transport. In combination with their expressed chemotaxis genes, we suggest that these motile cells enable cooperation in their local sub-community by attaching to surfaces or by forming aggregates.

Sporulation related genes are present in the MAGs KR46_Ju.mb.14, KR46_M_MetaG.mb.3, KR46_S_MetaG.mb.7, and MM_PW_MetaG.mb.36 out of the 17 MAGs/SAGs representatives of the phyla Firmicutes, Firmicutes_A, and Firmicutes_B (Supplementary Data 11). Sporulation might be a potential mechanism for these MAGs to cope with environmental stressors such as low carbon and energy conditions. However, this survival mechanism has been suggested to be a 'dead-end strategy' in the long-term due to e.g. the need for cell repair upon revival[49]. An additional possible detrimental aspect may include the response time to a transient nutrient and energy supply that may be consumed by active cells before the spore has germinated.

**Active but episodic microbial life in deep groundwater.** Many type II toxin-antitoxin systems (TA) prevalent in prokaryotes[50] are encoded/expressed in the reconstructed MAGs/SAGs of Fennoscandian Shield oligotrophic groundwaters (e.g., PemK-like, MazF-like, RelE/RelB, and BrnT/BrnA, etc.) that are likely to alleviate environmental stressors (Supplementary Data 12). The prevalence of TA system genes in FSGD MAGs/SAGs was greater compared to shallow aquifer MAGs (2402 MAGs from an aquifer adjacent to the Colorado River near Rifle, CO, USA[51]) (Supplementary Data 12). TA systems are envisioned to participate in a range of cellular processes such as gene regulation, growth arrest, sub-clonal persistence, and cell survival[52]. The envisioned role of these TA systems in the growth arrest in response to starvation is hypothesized to improve survival during starvation and help with the preservation of the common goods[52]. TA systems often fulfill their regulatory role by halting protein synthesis in response to environmental stimuli. We propose a theory in which deep groundwater microbes adjust to the very nutrient-poor conditions by TA systems triggering bacteriostasis to avoid exhausting the basal energy supply. To restart the cell function in the occasion of ephemeral access to nutrients, the autoregulation of the antitoxin component of the TA system is alleviated to defy the excess of toxin. We hypothesize that the reversible bacteriostasis imposed by TA systems could potentially help with sustaining life in the extremely oligotrophic deep groundwater ecosystems.

Additionally, we recovered error-prone DNA polymerase (DnaE2) in reconstructed FSGD MAGs/SAGs of 75 genome clusters (223 MAGs/SAGs) with Supplementary Fig. S7 showing the phylogeny of type-C polymerases including DnaE2 and a list of genomes containing DnaE2 are found in Supplementary Data 13. These polymerases can be recruited to stalled replication

forks and are known to be involved in error-prone DNA damage tolerance[53] helping with the genome replication and potentially facilitating cell restart after the initial halt enforced by the TA systems.

Life in deep groundwaters consisting chiefly of microbes[54] feature spatial heterogeneity in response to factors such as bedrock lithology, depth, and available electron acceptor and donors. Spatial heterogeneity together with limited accessibility have so far hindered our understanding of the ecological and evolutionary forces governing the colonization and propagation of microbes in the deep groundwater niches. Based on our high-resolution exploration of microbial communities in the disconnected fracture fluids running through similar lithologies at two separate locations, we propose the existence of a common core microbiome in these deep groundwaters. The metabolic context of this common core microbiome proves that both physical filters and biological interactions are involved in defining the dimensions of fixed niches where dissimilatory sulfate metabolism and reciprocal symbiotic partnerships seem to be among the most favored traits. By providing genomic and transcriptomic proof for the dormancy via reversible bacteriostatic functions such as TA systems, we expand the understanding of the ecological aspect of the microbial seed bank concept[55] in the deep oligotrophic groundwaters by suggesting an active but episodic strategy for the phylogenetically diverse microbiome of the deep groundwater in response to ephemeral nutrient pulses. We speculate that instead of a lifestyle where microbes predominantly invest in the functions related to maintenance, reversible bacteriostatic functions enable an episodic lifestyle that avoids exhausting the basal energy requirement. However, as these are genome-informed speculations that episodic lifestyles may play a significant role in the microbiomes of deep oligotrophic groundwaters, there is a need for more empirical work to confirm or refute these predictions.

## Methods

**Sampling and multi-omics analysis**. Multiple groundwater samples were collected over several years from two deep geological sites excavated in crystalline bedrock of the Fennoscandian Shield. The first is the Swedish Nuclear Fuel and Waste Management Company (SKB) operated Äspö HRL located in the southeast of Sweden (Lat N 57° 26' 4" Lon E 16° 39' 36"). The second site is on the island of Olkiluoto, Finland, which will host a deep geological repository for the final disposal of spent nuclear fuel (Lat N 61° 14' 31", Lon E 21° 29' 23"). Water-types with various ages and origins were targeted by sampling fracture fluids from different depths. The Äspö HRL samples originated from boreholes SA1229A-1 (171.3 mbsl), KA3105A-4 (415.2 mbsl), KA2198A (294.1 mbsl), KA3385A-1 (448.4 mbsl), and KF0069A01 (454.8 mbsl). The Olkiluoto samples originated from drillholes OL-KR11 (366.7–383.5 mbsl), OL-KR13 (330.5–337.9 mbsl), and OL-KR46 (528.7–531.5 mbsl). Detailed lithologies of these sites are described in the Supplementary Information.

Collected samples were subjected to high-resolution analysis by combining metagenomics ($n = 27$ from the Äspö HRL and $n = 17$ from Olkiluoto), single-cell genomics ($n = 564$), and metatranscriptomics ($n = 9$) (detailed information of the physicochemical characteristics of the samples and stats of the generated datasets are presented in Supplementary Data 1). Single-cell amplified genomes (SAGs) were captured from KA3105A-4 ($n = 15$), KA3385A-1 ($n = 148$), SA1229A-1 ($n = 118$), OL-KR11 ($n = 138$), OL-KR13 ($n = 117$), and OL-KR46 ($n = 28$) water samples. To probe the expression pattern of the resident community, metatranscriptomic datasets were generated for Äspö HRL samples[1,24] originating from boreholes KA3105A-4 ($n = 2$), KA3385A-1 ($n = 4$), and SA1229A-1 ($n = 3$). Details of sampling, filtration, DNA/RNA processing, and geochemical parameters of the water samples along with statistics of the metagenomics/metatranscriptomics datasets and SAGs are available in Supplementary Data 1 and Supplementary Information Document.

**Metagenome assembly**. All datasets were separately assembled using MEGAHIT[56] (v. 1.1.1 or v. 1.1.2 as specified in Supplementary Data 1) with settings–k-min 21–k-max 141–k-step 12–min-count 2. The datasets originating from the same water type in each location were also processed as co-assemblies in order to increase genome recovery rates (using the same assembly parameters). A complete list of all metagenomic datasets assembled in this study ($n = 44$) and the co-assemblies are provided in Supplementary Data 1.

**Fennoscandian shield genomic database (FSGD)**. The generated data were used to construct a comprehensive genomic and metatranscriptomic database of the extremely oligotrophic deep groundwaters. Automated binning was performed on assembled ≥ 2 kb contigs of each assembly using MetaBAT2[57] (v. 2.12.1) with default settings. Quality and completeness of the reconstructed MAGs and SAGs were estimated with CheckM[58] (v. 1.0.7). The taxonomy of MAGs/SAGs with ≥50% completeness ≤ 5% contamination was assigned using GTDB-tk[59] (v. 0.2.2) that identifies, aligns, and concatenates marker genes in genomes. GTDB-tk then uses these concatenated alignments to place the genomes (using pplacer[60]) into a curated reference tree with subsequent taxonomic classification. Phylogenomic trees of the archaeal and bacterial MAGs and SAGs were also created using the "denovo_wf" subcommand of GTDB-tk (–outgroup_taxon p_Patescibacteria) that utilizes FastTree[61] (v. 2.1.10) with parameters "-wag -gamma". Reconstructed MAGs and SAGs were de-replicated using fastANI[62] (v. 1.1) at ≥ 95% identity and ≥ 70% coverage thresholds. A detailed description and genome statistics of the Fennoscandian Shield genomic database (FSGD) is shown in Supplementary Data 2 and Supplementary Information.

**Functional analysis of the reconstructed genomes**. Annotation of function, validation of annotations, computation of isoelectric points and codon usage frequency[63], abundance, and expression analysis (metatranscriptome) are detailed in the Supplementary Information.

**Reporting summary**. Further information on research design is available in the Nature Research Reporting Summary linked to this article.

## Data availability

Metagenomes, SAGs, MAGs, and metatranscriptomes that support the findings of this study are deposited in GenBank and their respective accession numbers are provided in Supplementary Data 1. The FSGD MAGs are deposited in GenBank under the NCBI BioProject with the accession number PRJNA627556. The MAGs and SAGs generated in this study are publicly available in figshare under the project "Fennoscandian Shield genomic database (FSGD)" with the identifier https://doi.org/10.6084/m9. figshare.12170313. Alignments and phylogenetic trees that support the findings of this study are available in figshare under the project "Fennoscandian Shield genomic database (FSGD)" with the identifiers https://doi.org/10.6084/m9.figshare.14166650, https://doi. org/10.6084/m9.figshare.14166638, https://doi.org/10.6084/m9.figshare.13298513, and https://doi.org/10.6084/m9.figshare.12170310. All data supporting the findings of this paper are available within this paper and its supplementary material. All the programs used and the version and set thresholds are mentioned in the manuscript, supplementary information, and the reporting summary. Source data are provided with this paper.

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

## Acknowledgements

The work conducted by the U.S. Department of Energy Joint Genome Institute, a DOE Office of Science User Facility, is supported under Contract No. DE-AC02-05CH11231. The Swedish Research Council (contracts 2018-04311, 2017-04422, and 2014-4398) and The Swedish Nuclear Fuel and Waste Management Company (SKB) supported the study. M.D. thanks the Crafoord Foundation (contracts 20180599 and 20130557), the Nova Center for University Studies, Research and Development, and Familjen Hellmans Stiftelse for financial support. M.D. and D.S. thank the Carl Tryggers Foundation (grant KF16: 18) for financial support. S.B. and M.M. acknowledge financial support from the Swedish Research Council and Science for Life Laboratory. High-throughput sequencing was also carried out at the National Genomics Infrastructure hosted by the Science for Life Laboratory. Bioinformatics analyses were carried out utilizing the Uppsala Multi-disciplinary Center for Advanced Computational Science (UPPMAX) at Uppsala University (projects b2013127, SNIC 2019/3-22, and SNIC 2020/5-19) with support from a SciLifeLab-WABI bioinformatics grant. We would also like to thank Mats Åström for his comments on the Äspö HRL lithology. JS is financially supported by the Knut and Alice Wallenberg Foundation as part of the National Bioinformatics Infrastructure Sweden at SciLifeLab.

## Author contributions

M.D., S.B., and R.B.-L. devised the study; M.L.-F. and E.B. collected and processed the samples; M.M., J.S., D.S., and M. B. analyzed the data. M.M. and M.D. interpreted the data and drafted the paper, and all authors read and approved the final paper.

## Funding

## Competing interests

The authors declare no competing interests.
