## [Peer Review File · Nature Communications]

Reviewers' Comments:

Reviewer #1:

Remarks to the Author:

In this manuscript, Mehrshad et al. used a multi-omics approach (transcriptomics, metagenomics, single cell genomics) to better understand how the microorganisms inhabiting in oligotrophic groundwater boreholes from the Fennoscandian Shield live. Their main findings are: (1) microbial community composition is correlated to the lithology of the borehole, (2) microorganisms living in these boreholes have a highly cooperative lifestyle, which is especially true through the formation of biofilms and aggregates. They discuss that this cooperative lifestyle is helping microorganisms alleviate the problem of the "tragedy of the common goods", an economics problem in which all users deplete the shared resource through their collective action. They propose a new model, called "halt and catch fire", where microorganisms cooperate with each other instead of relying solely on maintenance and survival in times of low nutrients.

The manuscript is overall well written and the figures are generally clear and well presented. A main strength of the manuscript is the quality and tremendous amount of data that were collected and analyzed. However, the strength of the manuscript also becomes their Achilles' heel, therefore their weakest point as the interpretation of the data is at times confusing, the conclusions are too far reaching, and their model should be better explained, as detailed below.

- The introduction is very short and does not reflect the body of knowledge we have on the deep crust biosphere. Although not many omics have been performed, we are getting more and more information about microorganisms that live in oligotrophic ground water (for example Bamfield, Moser, Onstott labs work, just to name a few).
- Some of the discussion is hard to follow and push the conclusions too far. For example, lines 189-196. The discussion about the potential role of viruses is not supported by the data. To be having such discussion about viruses, it could be worth incorporating a section where they discuss CRISPRs (for more than one cluster) and look for viral sequences and the types of viruses in the metagenomes and metatranscriptomes.
- The authors make comparisons using the "tragedy of common goods" (which technically is called "tragedy of the commons") (lines 207-214). Because of that, they use many words in quotations, which makes the text confusing to read. For example, it becomes unclear who the "cheaters" would be in the underground microbial community. While the idea of the economic comparison is good, it is not fully integrated in the text, leaving the reader confused. The manuscript solely focus on the analysis of the genomic data (SAGs, MAGs, metatranscriptomes), but discuss models that would greatly benefit from the geochemical data as well.
- It is unclear how they go from a high prevalence of Patescibacteriota and Nanoarchaeota to how they are dependent on the community and live in cooperation.
- The authors propose a model called "Halt and Catch Fire". While catchy, the name of the model is misleading as 1) the computer idiom is based on the fact that you need the computer ceased meaningful operations and restart the computer. While cells may be halting their growth, there was no indication of this in their transcriptomic data. Moreover, there was also no indication that there was anything "catching on fire", therefore high expression rates since they did not measure rates over time; 2) the model do not fit with their cooperation theory. The model is mentioned in the last sentence of the manuscript with no clear explanation or description. It would have been interesting to have a paragraph on it, and even a figure, to explain what they mean. In a system that "halts", one could assume that many microorganisms can undergo sporulation or other forms of dormancy, but there is no such discussion in the manuscript.

Minor notes:

The authors keep referring to the "tragedy of common goods", but the expression is "tragedy of the commons".

Line 116: Are data available regarding the available reducing agents and organic carbon? If so, need to cite the publication, and possibly add a table to the material and methods.

Lines 123-124: When referring to the domain, Bacteria and Archaea should be capitalized.

Line 129: The lithologies should be discussed or referred to a previous paper.

Lines 150-152: Need to refer to table.

Figure 1 B and C: The figure may be hard to view for color-blind people.

Supplementary Figure 4: Instead of arbitrarily assigning a root, the tree should be unrooted.

Reviewer #2:

Remarks to the Author:

The manuscript "Energy efficiency and biological interactions define the core microbiome of deep oligotrophic groundwater" by Mehrshad et al deals with a "multi-omic" analysis of deep groundwater microbiomes from two Scandinavian sampling sites. They use genome-resolved metagenomics and single cell genomics to create a genome database of microbes in these (partially) time-resolved ecosystems and add metatranscriptomics to understand the activity of the microbial communities in the system and create an ecosystem understanding.

Although the data is extremely promising and a good asset to the deep subsurface community, I do have some reservations regarding this manuscript (major issues):

- The major messages of the manuscript are based on speculations or conclusions. This is very prominent in the abstract, which does not state a single result that the authors generated but rather is an accumulation of conclusions and speculations. For instance, the authors state in line 32 and 33 "The fitness of primary energy producers is increased by ecological traits such as aggregate or biofilm formation." The authors have not measured fitness or primary production, they have not measured aggregate or biofilm formation, yet they build causality between these objects. Another example is the fact, that the authors sampled two sites and draw conclusions for the entire deep biosphere on Earth. The authors have, however, not compared their sites to other deep biosphere ecosystems with regard to cell counts, diversity and water composition (which I'm requesting throughout the review to do). This major issue of building the manuscript based on conclusions or speculations holds true for the most part of the it and I've highlighted other examples in the minor comments. The manuscript would need a complete re-writing in that regard and be drastically toned down.

- Solid statistics are missing where authors draw conclusions from their findings. For instance, they state that they find DnaE2 to be expressed in MAGs but they do not state for how many. Further, they draw very big conclusions about the specific function of these proteins and propose a new paradigm "Halt and catch fire" without knowing if the catalytic centers and the necessary domains are conserved in the proteins they look at. This is just one example of multiple analyses that are missing statistics and proper in-depth analyses.

- The study is missing cell counts on the groundwater samples. This is a standard technique that is done for every groundwater sampling. Please add that data for each metagenomic or metatranscriptomic sample and show that your samples are representative of a deep biosphere (compare to <https://www.nature.com/articles/s41561-018-0221-6>). Due to the fact that the authors were able to perform metatranscriptomics I assume that the cell count was fairly high and is thus not representative for a deep groundwater community in general. The authors also state that the sampled groundwaters are of the same characteristics but do not provide the data. Please provide gaseous composition and the major anion and cation concentrations for each site and their stability over time. Finally, relate the stability to the respective microbiome analyses because they could be responsible for some of the changes.

Minor comments:

- L 24: Euks are only seldomly recovered in deep groundwaters. Also the mentioning of viruses does not help the story because neither Euks nor viruses are analyzed in this manuscript. I suggest narrowing the focus to the real question here.

- L 24: habitat is restricted to one organisms, please use ecosystem instead (here and elsewhere

in the manuscript)

- L 26: First sentence does not include which methods the authors used. This comes up vaguely in L 29 as "multi-omics" which is sort of misleading as they used genomics and transcriptomics, two methods don't make it "multi". Focus on content rather than on buzz words. This phrase here doesn't help the reader at all, it's rather frustrating.
- L 30: "understanding of the underlying mechanisms" – of what? This sounds rather like a commonly used phrase but it doesn't have content for the reader.
- L 32/33: Entire statement is speculation. There is no data for primary production, no data for fitness, no data for biofilm formation, no data for aggregate formation.
- L 33: there is no data for partnerships in this manuscript. There is no data showing that the analyzed microbes are epi-bionts.
- L 33/34: although I have done a lot of research in groundwater microbiology including Patescibacteria, I do not understand the content of this sentence without having read the entire manuscript before.
- L 35: You say "invest" but in fact you mean that you have measured transcripts for respective genes. This is misleading on so many levels and does not reflect the science that you did.
- L 39: You mention food webs of deep groundwater without any citation for this statement. To the best of my knowledge nobody has ever shown how carbon fluxes work in groundwater and nobody knows if there actually is a food chain (or if all of the microbiome is just a remnant from sessile microbes?).
- L 40: Euks are rather an exception in the deep subsurface. I'd suggest to rephrase unless you show that this is really the case for your system as well.
- L 42: The deep groundwater ecosystems are not one environment. These are many. Please rephrase.
- L 45: There is no such thing as one deep core groundwater microbiome. It varies across the entire planet. Only functions are conserved as in almost all ecosystems that are similar.
- L 47: The authors state again that the deep disconnected biosphere is one environment. This is simply wrong.
- L 57/58: Habitats cannot lack a method. Consider rephrasing.
- L 60: please elaborate on how these tunnels might influence the deep biosphere that you sample (e.g., via oxygen).
- L 65: Is there one biome in these different sampling sites? That might be a stretch to state in the introduction.
- L 64 – 68: The authors study two sites but draw conclusions for the entire deep biosphere. The manuscript needs to be toned down drastically.
- L 75: Where are the data on the characteristics of the groundwater? Stating that they have different characteristics without showing them is not scientific. Also add statistical tests and replicates to show the differences. Please compare the characteristics to other study sites to show that your groundwater is representative for the majority of the deep biosphere.
- L 80: Were the MAGs curated regarding GC and coverage of the single scaffolds (in, e.g., ANVIO)? The authors should do that as there might be many biases in the MAGs otherwise (see <https://www.nature.com/articles/s41564-018-0171-1>)
- L 93: please compare your diversity to the diversity that other studies found. Is your data representative of the deep biosphere?
- L 104: It is problematic to map reads against all created population genomes as there might be redundancy. The authors should de-replicate their genome dataset based on ANI.
- L 119: The authors suddenly talk about size fractionation. Please explain the rationale behind the study setup and how this fractionation impacted your sample normalization strategy (which I assume has been done based on number of sequenced base-pairs?).
- L 121: The authors determine the presence/absence of a genome based on mapping coverage. However, mapping coverage is meaningless here. The authors should calculate how much of the genome is actually covered by reads rather than how many reads have mapped (many reads can map in one region leaving other regions uncovered).
- L 132: You make a general statement here but you only show this for two sites. Please rephrase / tone down.
- L 135: If something doesn't confute a possibility it cannot be used as evidence but that's how you argue. Please do provide solid evidence for evolutionary convergence between the sites.
- L 135: Please demonstrate that the two sites were not connected in the past (elaborate on geology / geological history of the sites).

- L137-139: Statistics missing.
- L 141: Please elaborate how you determine that these are evolutionary timescales.
- L 142: You claim that basic proteins are enriched in one sample. Please provide statistical tests for this.
- L 145: You claim the two samples are similar. Provide a statistical test (e.g., Power test).
- L 148: Species richness cannot be determined from binned genomes. Maybe that one sample with low richness was problematic in good binning? Please add an analysis of species richness of all samples based on marker genes. Do not use ribosomal RNA for that as it doesn't get assembled well. Use a house keeping gene that is good for species delineation, e.g. *gyrB*
- L 161: "these groundwaters" -- do these references refer to the sites that you sampled here as well? If so, please perform a comparison of the respective findings! If not, please rephrase.
- L 169: Sentence ending with citations 22/23. Please elaborate more on why this is the case otherwise the reader won't understand your message.
- L 188: Not only viruses! CRISPR-Cas systems can work against all mobile genetic elements.
- L 190: use past tense "expressed"
- L 194-196: pure speculation, no evidence for this.
- L 216: Using the term Nanoarchaeota from GTDB is extremely misleading. I suggest using a term that does not have a historic connection to a specific organism group, like Woese archaeota etc.
- L 219: Please provide evidence for leaky metabolites (metabolomics) and elaborate more or remove the statement. This has not been shown before and is a huge claim.
- L 223: Lipid and nucleotide metabolism is usually sparse in Patescibacteria. Are these binning errors since you didn't manually check all the MAGs? Please elaborate more.
- L 236: I do not agree with the conclusion that your data underscores cooperation as a competent evolutionary strategy. The easiest answer is, that the episymbionts can have multiple hosts or do not rely on hosts at all and cooperation is not relevant, rather cell death.
- L 281: Since these genes build the core of your manuscript, please elaborate how many genomes of the community possess these genes! Add statistics.
- L 281: Please perform a phylogeny as well as a motif search for the active center of the recovered proteins to demonstrate that they are conserved and are likely performing the predicted function. Modelling of the protein 3D structure could demonstrate the robustness of your results.
- L 284: Period missing.
- The authors should make all genomic and transcriptomic data publicly available.

Reviewer #3:

Remarks to the Author:

This manuscript by Mehrshad/Lopez-Fernandez et al studies the microbiome of the deep subsurface and describes the ecology of the organisms. Using metagenomic and metatranscriptomic analyses, the authors determine the abundance, activity, and ecology of various taxa. This paper presents a large and interesting dataset, adding a significant number of new genomes from distinct and diverse taxa. As a data resource, it is extremely valuable and could lead to many follow-up analyses by researchers studying particular taxa, pathways, and ecosystems.

I was very excited to read this manuscript because it describes a fascinating system with enormous potential for discovery. The systems described are amongst the deepest terrestrial systems that have been studied, and certainly no other deep terrestrial subsurface ecosystem has been studied to the same depth from the perspective of microbiology. Certainly, this provides tremendous opportunities to answer the kind of broad ecological and evolutionary questions such as those proposed here.

Overall, I found this paper to be somewhat underwhelming. While the genomes and transcriptomes presented here will certainly be useful as a phylogenetically broad data resource – the emphasis is largely on the taxon-specific gene/protein content of MAGs and SAGs from across Bacteria and Archaea, especially epibiont lineages such as the Patescibacteria and Nanoarchaeota. I found the connection to the ecosystem – which really drives the paper's significance, in my opinion – to be rather rudimentary. Contrary to the stated objectives and conclusions, I found little

synthesis/integration of interactions, energy efficiency, and omics data. At the end of the paper, it was not clear to me what new understanding was achieved for this system – though clearly there are many new genomes and taxa. I really liked the analyses on codon usage and isoelectric points. “Halt and catch fire” is an exciting and catchy name but the paper does not present substantive analyses to justify that microorganisms pursue this strategy. I also commend the authors on making all of their data public and well written and easy to follow methods.

In summary in its current form, this paper seems more appropriate as a data resource; otherwise significant efforts will be required to attain a more ecological/evolutionary perspective and to better support the conclusions with the Figures and Results.

Major concerns

1. A fundamental finding is presented in the form of the core microbiome for deep oligotrophic groundwaters. But this is never defined in the study. Which members constitute the core microbiome and what is the extent of overlap of their niches? Supplementary Table 4 contains a lot of raw data, but the authors should clearly define the “core” microbiome instead of letting the reader interpret this.
2. A second fundamental finding is presented in the form of “description of ecological and evolutionary processes essential for successfully occupying and propagating in the available niches.”. Again, I found this underwhelming. While a number of the findings such as the potential regulation of transcription are intriguing – there is currently insufficient evidence presented. Some comparisons and potentially answering questions such as “Are the number of Toxin/Anti toxin systems greater than one would typically observe in other environments like shallow aquifers?” would bolster this argument.
3. Lines 194-196 It is unclear to be as to why this would be associated with phages defense. Could the BrnT/BrnA toxin-antitoxin system simply have been active in these cells at the same time?
4. In a highly energy-limited environment, it is understandable that there may be significant metabolic interactions between organisms, specifically Patescibacteria and Nanoarchaeota. However, some analyses on specific associations (even predicted) of these organisms with their hosts would constitute a significant advance.
5. Metabolic analyses are not comprehensive. Sulfur species (specifically sulfate) is presented as an important electron acceptor in these systems. Yet, no comprehensive analyses were conducted to verify the metabolic potential for sulfate reduction. Given the reversible nature of pathways associated in sulfate reduction, it is important to conduct these analyses to verify the claims in the study.
6. This study seems to be presented in a vacuum. It seems that the authors have recently published similar papers on sulfur cycling and metatranscriptomes from these environments. Some background demonstrating the advances made by this current study and contrasts with the previous study will provide some useful context.
7. Figure 6 is not present in the manuscript. I have gone over the individual files as well and cannot find this.
8. Line 336 Is the coverage threshold accurate?

REVIEWER COMMENTS

Reviewer #1

C1- The manuscript is overall well written and the figures are generally clear and well presented. A main strength of the manuscript is the quality and tremendous amount of data that were collected and analyzed. However, the strength of the manuscript also becomes their Achille's heel, therefore their weakest point as the interpretation of the data is at times confusing, the conclusions are too far reaching, and their model should be better explained, as detailed below.

We would like to thank the reviewer for the overall positive assessment.

C2- The introduction is very short and does not reflect the body of knowledge we have on the deep crust biosphere. Although not many omics have been performed, we are getting more and more information about microorganisms that live in oligotrophic ground water (for example Bamfield, Moser, Onstott labs work, just to name a few).

To emphasize the state of our knowledge on groundwaters and specifically from the genomic perspective, we have compiled a list of publicly available metagenomic datasets for groundwater and added information regarding the geographical location, depth, and amount of sequenced data as Supplementary Figure S1. Additionally, the well-studied and large-scale genome-resolved studies of groundwater are mentioned in the introduction.

The microbial diversity of the terrestrial subsurface has previously been probed by large-scale “omics” in shallow aquifers (Rifle, USA)², a CO₂ saturated geyser (Crystal Geyser, Utah, USA)³, and carbon-rich shales (Marcellus and Utica, USA)⁴. However, the microbiomes of extremely oligotrophic and disconnected deep groundwater ecosystems is still missing a comprehensive comparative “multi-omics” analysis.

As per reviewers’ suggestion, we have now also added the requested context and related references for the existing publications studying deep oligotrophic groundwaters in the supplementary information document under the headline “**Background**” and referred to this in the introduction section. For convenience, we also include this section below.

Background

During the last decades, it has been demonstrated that microorganisms inhabit the sub-surface biome down to several kilometers below the surface. Although microbial life in the deep crust biosphere has been gaining interest, due to its difficulty to access it is still one of the least understood environments on earth. Consequently, many novel taxa are being identified in the deep biosphere^{1,5}.

The continental deep biosphere is estimated to contain 2 to 6 × 10²⁹ cells^{6,7}, containing members from all three domains of life along with viruses^{4,8-13}. Even though the available energy flux is very low compared to that of the photosynthetically fixed carbon on the surface¹⁴, microorganisms are widely spread in oceanic crust fluids¹⁵⁻¹⁷, marine sediments^{18,19}, terrestrial rocks^{20,21}, and granitic groundwaters²²⁻²⁴. In addition, many of these deep biosphere microbes are both alive and active^{10,25-29}.

In continental groundwaters, microbial activity is strongly positively correlated to the proximity of the photosynthesis-fueled surface⁷ and thus, water-bearing deep fracture systems are extremely oligotrophic³⁰. A recent study showed that there is a steady flow of surface organisms to the deep subsurface with a selection event resulting in some taxa adapting to the new conditions while others perish²³. In addition, Lopez-Fernandez et al.²⁷ showed the presence of viable taxa in a deep continental crystalline rock and that any non-viable cells are rapidly degraded and recycled into new biomass²⁷. The deep biosphere is suggested to be adapted to the low energy conditions by e.g. small cell size and streamlined genomes³¹. Microbial populations with the potential to initiate biofilm formation were also identified in deep terrestrial subsurface waters¹¹. This close proximity likely promotes syntrophy and cycling of nutrients between populations³².

C3- Some of the discussion is hard to follow and push the conclusions too far. For example, lines 189-196. The discussion about the potential role of viruses is not supported by the data. To be having such discussion about viruses, it could be worth incorporating a section where they discuss CRISPRs (for more

than one cluster) and look for viral sequences and the types of viruses in the metagenomes and metatranscriptomes.

We agree with the reviewer that viruses are a very interesting component of life in the deep groundwater. However, analyzing viruses in our metagenomic and metatranscriptomic datasets is beyond the main point and scope of this manuscript. We would like to mention that we are currently exploring viruses and their potential ecological roles in the deep terrestrial biosphere in another manuscript. We believe that this work will provide additional perspectives on life in these groundwater ecosystems and be complementary to the current manuscript.

For the section that the reviewer is referring to, the main focus is different codon usage patterns between the expressed genes and the rest of the genome that is evident based on codon frequency analyses and also inspection of functions of these genes supports the potential impact of codon usage bias in tuning the expression pattern. We have now rephrased this paragraph to make sure this message is conveyed more clearly. The detailed annotations of these expressed genes are also presented in the Supplementary Table S5.

C4- The authors make comparisons using the “tragedy of common goods” (which technically is called “tragedy of the commons”) (lines 207-214). Because of that, they use many words in quotations, which makes the text confusing to read. For example, it becomes unclear who the “cheaters” would be in the underground microbial community. While the idea of the economic comparison is good, it is not fully integrated in the text, leaving the reader confused. The manuscript solely focus on the analysis of the genomic data (SAGs, MAGs, metatranscriptomes), but discuss models that would greatly benefit from the geochemical data as well.

We have now used the “tragedy of the commons” phrase throughout the text and thank the reviewer for pointing this out.

In the deep groundwater ecosystem, any group that is profiting on public goods without contributing are defined as cheaters as they would be in any other ecosystem. Examples of these groups in the deep groundwaters could be Patescibacteria and DPANN archaea or any other group that is using the produced public good. As it is shown in figure 6, a considerable part of the expressed genes in the metatranscriptomes belong to the potential public goods that could be potentially exploited by the cheaters.

We agree with the reviewer that the addition of geochemistry will add to the robustness of our analysis and include key parameters in Supplementary Table S1. We here focus on the metabolic potential of reconstructed MAGs and SAGs and therefore incorporated the geochemistry dimension by analyzing the dissimilatory metabolisms of reconstructed MAGs and SAGs belonging to the active members of the community. These results are in line with geochemistry information at large for example sulfate is present in all of the groundwaters where sulfur metabolism is predicted (Supplementary Table S1). By inspecting the genomic potential of these microbes, we get a frame by frame view of the mechanisms of slow-motion life in the deep groundwater. As previously shown in more detail using proteomics data, extensive analysis of the genomic potential of cells can unravel an active sulfur cycling that remain undetectable by geochemistry analysis alone³². As microbes are the engines that support the major part of biogeochemistry in these ecosystems, we believe the extensive analysis of their genomic potential and overlaying the results of metatranscriptomics to target the most active groups provide highly useful and valuable information on prevalent modes of life and types of biological interactions in these ecosystems.

C5- It is unclear how they go from a high prevalence of Patescibacteriota and Nanoarchaeota to how they are dependent on the community and live in cooperation.

We have now rephrased this part to make it clearer (Lines 240-273).

C6- The authors propose a model called “Halt and Catch Fire”. While catchy, the name of the model is misleading as 1) the computer idiom is based on the fact that you need the computer ceased meaningful operations and restart the computer. While cells may be halting their growth, there was no indication of this in their transcriptomic data. Moreover, there was also no indication that there was anything “catching on fire”, therefore high expression rates since they did not measure rates over time; 2) the model do not fit with their cooperation theory. The model is mentioned in the last sentence of the manuscript with no clear explanation or description. It would have been interesting to have a paragraph on it, and even a figure, to explain what they mean. In a system that “halts”, one could assume that many microorganisms can undergo sporulation or other forms of dormancy, but there is no such discussion in the manuscript.

We have now added detailed annotations regarding the sporulation to the discussion. Sporulation related genes are partially present in MAGs KR46_Ju.mb.14, KR46_M_MetaG.mb.3, KR46_S_MetaG.mb.7, and MM_PW_MetaG.mb.36 among the 17 MAGs/SAGs representatives of the phyla Firmicutes, Firmicutes_A, and Firmicutes_B (Supplementary Table S10). Sporulation might be the potential coping mechanism of these MAGs in response to environmental stressors. However, this survival mechanism may prove detrimental in the long-term due to the response time to a transient nutrient and energy supply that may be consumed by active cells before the spore has germinated.

We believe that not every cell will follow the same strategy to cope with these extreme ecosystems. We also explore other reversible bacteriostatic processes such as those enabled by toxin-antitoxin systems in response to environmental stressors and suggest that those who have these genetic circuit could potentially leverage it to avoid exhausting basal energy requirements.

We have now expanded the section explaining the proposed halt and catch fire mode of life. We also included a comparative analysis with 2402 shallow aquifer MAGs and our results show that a higher percentage of FSGD MAGs/SAGs contain toxin-antitoxin systems. The HCF opcode is functioning as a reversible halt of function in the computer sciences. This way the computer CPU will not get exhausted and the process will be stopped before causing irreversible damage to the hardware. MAGs/SAGs generated from these deep oligotrophic ecosystems also use these reversible bacteriostatic approaches to avoid exhausting the cell’s basal energy requirements. In the text we have examples of such functions being expressed as well. Additionally, the suggested mode of life matches cooperation among cells. For example, one proposed function of the reversible bacteriostatic function of the toxin/antitoxins is to avoid depletion of common goods that could potentially be a cooperative act at the community level. While our genome-resolved multi-omics analyses supports this model, more empirical studies in future are needed to further explore the modes of life employed by deep oligotrophic groundwater microbiome.

C7- The authors keep referring to the “tragedy of common goods”, but the expression is “tragedy of the commons”.

Thank you for pointing this out, we have now edited the text and used the phrase “tragedy of the commons” throughout the text.

C8- Line 116: Are data available regarding the available reducing agents and organic carbon? If so, need to cite the publication, and possible add a table to the material and methods.

A description of the metagenomic datasets and the physicochemical features of their samples is provided in Supplementary Table S1. For clarity, we have also added a sentence referring to this content in the methods section.

C9- Lines 123-124: When referring to the domain, Bacteria and Archaea should be capitalized. We have altered the text as suggested by the reviewer.

C10- Line 129: The lithologies should be discussed or referred to a previous paper.

In the introduction section (L58-60), we refer to several peer reviewed publications and site description documentations published by the operating organizations for both sites and also describe the sites and samples used in this study in the supplementary information document and Supplementary Table S1. As per reviewer's suggestion, we have now added a section describing the lithologies of these two sites to the supplementary information document under the headline "Site lithologies" and included more references there. Added text:

Site lithologies

The two sampling locations, Äspö HRL and Olkiluoto Island drillholes, are situated on opposite sides of the Baltic Sea on the Swedish (Lat N 57° 26' 4" Lon E 16° 39' 36") and Finnish (Lat N 61° 14' 31", Lon E 21° 29' 23") coasts. The crystalline bedrock of Sweden and Finland is part of the Precambrian Fennoscandian Shield that is predominantly made up of granite and quartz monzodiorite of quartz and aluminosilicate minerals including mica and feldspar.

The bedrock in the Äspö HRL region consists of overall well preserved (locally low-grade metamorphism and discrete foliation occur) Palaeoproterozoic ~1.8 Ga granitoids of the Transscandinavian Igneous Belt³¹. At the Äspö HRL site, these rocks have a composition ranging from diorite to granite³². The rocks are also cut by a number of deformation zones and open fractures, which frequently have surfaces covered by high and/or low temperature 1-15 mm thick secondary-mineral precipitates including calcite, chlorite, pyrite, clay minerals, epidote, adularia, and hematite³³. Hence, the fluids flowing in the fractures are in direct contact with both Precambrian granitoids and a variety of secondary minerals of variable age.

Olkiluoto is located in the southern Satakunta region in south-western Finland. The ~1.8 Ga Palaeoproterozoic bedrock of the southern Satakunta region is composed of supracrustal, metasedimentary and metavolcanics rocks deformed and metamorphosed during the Svecofennian orogeny³⁴. The main rock types at Olkiluoto are mica and veined gneisses, migmatite granite, grey gneisses and diabase. Minor veins and dykes are quartz feldspar gneisses and amphibolites. Mica and veined gneisses contain calcite, sulphides and clay mineral fracture fillings³⁵.

C11- Lines 150-152: Need to refer to table.

We have altered the text as suggested by the reviewer.

C12- Figure 1 B and C: Th figure may be hard to view for color-blind people.

Thank you for pointing this out and we have now edited the figure using a color-blind compatible palette.

C13- Supplementary Figure 4: Instead of arbitrarily assigning a root, the tree should be unrooted.

This tree was constructed by referring to the Timinskas et al.³³ publication where they analyzed different types of DNA polymerase III subunit α and its phylogeny. The alignment used for reconstructing this phylogeny is publicly available via figshare at <https://doi.org/10.6084/m9.figshare.13298513.v1>. Following their approach, we have used the same arbitrary rooting and believe that this is a valid approach. As per reviewer's suggestion, we have also uploaded the unrooted tree to figshare and the links for these online contents have been added to the figure legend.

Reviewer #2

C14- The major messages of the manuscript are based on speculations or conclusions. This is very prominent in the abstract, which does not state a single result that the authors generated but rather is

an accumulation of conclusions and speculations. For instance, the authors state in line 32 and 33 “The fitness of primary energy producers is increased by ecological traits such as aggregate or biofilm formation.” The authors have not measured fitness or primary production, they have not measured aggregate or biofilm formation, yet they build causality between these objects. Another example is the fact, that the authors sampled two sites and draw conclusions for the entire deep biosphere on Earth. The authors have, however, not compared their sites to other deep biosphere ecosystems with regard to cell counts, diversity and water composition (which I’m requesting throughout the review to do). This major issue of building the manuscript based on conclusions or speculations holds true for the most part of the it and I’ve highlighted other examples in the minor comments. The manuscript would need a complete re-writing in that regard and be drastically toned down.

As suggested, we have rewritten the abstract to include and focus on our results and refrain from extensive speculation. We also limit our conclusions to the sampled sites in this study and rephrased these sections to make sure this is conveyed clearly. Cell counts and other physicochemical characteristics of the samples are presented in Supplementary Table S1.

We have now analyzed the species richness and diversity based of gene *gyrB* as the reviewer suggested and included these results in the manuscript.

We have also now compared our FSGD genomic database with 2402 MAGs generated by Anantharaman et al.¹ from a shallow aquifer adjacent to the Colorado River near Rifle, CO, USA and have attempted to address your suggestions by means of these new analyses.

C15- Solid statistics are missing where authors draw conclusions from their findings. For instance, they state that they find DnaE2 to be expressed in MAGs but they do not state for how many. Further, they draw very big conclusions about the specific function of these proteins and propose a new paradigm “Halt and catch fire” without knowing if the catalytic centers and the necessary domains are conserved in the proteins they look at. This is just one example of multiple analyses that are missing statistics and proper in-depth analyses.

We have now also mentioned the number of genomes containing the DnaE2 gene in the text and the list of these MAGs/SAGs is added as the Supplementary Table S12.

We have now added the requested statistical tests as explained in the point-by-point answers to the comments.

All gene annotations were thoroughly checked by inspecting the conserved domains and for the DnaE2 gene specifically, we generated a phylogeny as represented in Supplementary Figure S7. We have now also included phylogeny reconstruction of other metabolic genes in the supplementary information.

C16- The study is missing cell counts on the groundwater samples. This is a standard technique that is done for every groundwater sampling. Please add that data for each metagenomic or metatranscriptomic sample and show that your samples are representative of a deep biosphere (compare to <https://www.nature.com/articles/s41561-018-0221-6>). Due to the fact that the authors were able to perform metatranscriptomics I assume that the cell count was fairly high and is thus not representative for a deep groundwater community in general.

The information for cell counts per milliliter of sample is presented in column AE of the Supplementary Table S1. Other information for the collected samples are also present in this table including (depth (mbsl), pH, water residence time, DOC (mg/L), $\delta^{18}\text{O}$, temperature, HCO^- (mg/L), NH^+ (mg/L), Fe^{2+} (mg/L), HS^- (mg/L), SO_4^{2-} (mg/L), Cl^- (mg/L), salinity (ppt), NO_2^- , NO_3^- , PO_4^{3-} , cell number (cell/ml), filter pore size, DNA extraction method, as well as dataset accession number, metagenome size Gb, and number of reads.

In the same supplementary table in the tab “metatranscriptome” we present information regarding the samples collected for the metatranscriptome sequencing. In the column “Total water volume (L)” we have included the information regarding the amount of water we filtered to collect the metatranscriptomes. To generate these metatranscriptomes, between 140 to 635 L water were concentrated and independent biological replicates for each borehole were generated. The amount of RNA and cDNA extracted for each metatranscriptome is also provided in this table. The method description and all the references are mentioned in the supplementary information document under the headline “RNA extraction and metatranscriptome sequencing”. The supplementary information also details the careful controls that were performed to show the validity of the data that was further confirmed by mapping of the metatranscriptomes to MAGs and SAGs reconstructed from these boreholes.

New low-input library preparation protocols allow for sequencing low biomass samples such as our groundwater samples and these methods are sufficiently sensitive for single-cell RNA sequencing. Considering this, obtaining sufficient RNA for sequencing from low biomass habitats is becoming more possible specifically in this case when large volumes of water were collected. These sites have been studied by the authors for approximately eight years from biological perspectives reflected in many peer-reviewed publications. All these studies and controls for our own samples confirm the origin of the collected samples from deep groundwater.

C17- The authors also state that the sampled groundwaters are of the same characteristics but do not provide the data. Please provide gaseous composition and the major anion and cation concentrations for each site and their stability over time. Finally, relate the stability to the respective microbiome analyses because they could be responsible for some of the changes.

Physicochemical characteristics are provided in Supplementary Table S1 for all collected samples. In the site description included in the supplemental information, we state that the groundwaters flow towards the boreholes and drillholes and in the case of the Äspö HRL, this causes mixing of the groundwaters (e.g. the thoroughly mixed water ‘TM-448.4’ analyzed in this study). The stability of the groundwaters over time for several of the Äspö HRL boreholes can be found in Lopez-Fernandez et al.²⁴ while chemical parameters were monitored during sampling at Olkiluoto to ensure that the water was representative of the isolated fracture.

We agree with the reviewer that these changes in geochemistry can cause differences in the microbial communities and state that we compare “groundwaters flowing in similar lithologies” that nevertheless “offer fixed niches that are occupied by a common core microbiome”.

C18-L 24: Euks are only seldomly recovered in deep groundwaters. Also the mentioning of viruses does not help the story because neither Euks nor viruses are analyzed in this manuscript. I suggest narrowing the focus to the real question here.

As per the reviewer’s suggestion, we have rephrased this section and rewrote major parts of the abstract.

C19-L 24: habitat is restricted to one organisms, please use ecosystem instead (here and elsewhere in the manuscript)

This has been replaced with other alternative words according to the context throughout the text.

C20-L 26: First sentence does not include which methods the authors used. This comes up vaguely in L 29 as “multi-omics” which is sort of misleading as they used genomics and transcriptomics, two methods don’t make it “multi”. Focus on content rather than on buzz words. This phrase here doesn’t help the reader at all, it’s rather frustrating.

In lines 26 to 32 we are describing the outlook of this study. We first mention the approach used in this study that is “genome-resolved comparative analysis” and in the following sentences describe the methods used for this comparative analysis. In response to the reviewer, we have rephrased this section to mention different methods earlier in the text.

We use this approach to provide a high-resolution description of the community and combine metagenomics, single-cell genomics, and metatranscriptomics to get a better understanding of the microbial community in the ecosystems studied here. Each of these methods provide a unique perspective and their combination allowed for a polyphasic and comprehensive analysis of the microbial community at a resolution unprecedented for deep oligotrophic groundwater. From a methodology perspective, metagenomics and single-cell genomics are two quite different omics methods with different outputs and resolution. While we understand that the reviewer is putting these two methods in the genomics category, we would prefer to keep them separated. The Venn diagram in figure 1A of our study shows the complementary nature of these methods in recovering genomes from deep groundwater.

C21-L 30: “understanding of the underlying mechanisms” – of what? This sounds rather like a commonly used phrase but it doesn’t have content for the reader.

We have now rephrased this sentence.

C22-L 32/33: Entire statement is speculation. There is no data for primary production, no data for fitness, no data for biofilm formation, no data for aggregate formation.

We have rewritten the abstract and mentioned more results.

C23-L 33: there is no data for partnerships in this manuscript. There is no data showing that the analyzed microbes are epi-bionts.

We have rephrased the text to highlight that the CPR Bacteria and DPANN Archaea MAGs/SAGs of this study lack core biosynthesis pathways for fatty acid based lipids and nucleotides. This and many prior reports based on reconstructed genomes of these groups also bring forward the possibility of symbiotic relations of these groups. Despite their enormous diversity, many of these taxa share features like having small cell sizes, reduced genomes, and limited biosynthetic capabilities. Consequently, these cells are expected to be dependent on their adjacent cells to supply them with metabolites they cannot synthesis themselves^{34–36}. The epi-symbiotic association of Patescibacteriota and Nanoarchaeota with prokaryotic hosts has already been verified for several representatives^{9,37–40} along with the recent work showing the very interesting parasitic interaction of *Vampirococcus* with its host via predation⁴¹. However, the level and range of host/partner specificity or the type of interactions for these different associations remain understudied. We have now changed ‘epibiont’ to symbiotic interactions that we envision for our MAGs/SAGs based on their metabolic deficiencies.

C24-L 33/34: although I have done a lot of research in groundwater microbiology including Patescibacteria, I do not understand the content of this sentence without having read the entire manuscript before.

We have now rephrased this sentence.

C25-L 35: You say “invest” but in fact you mean that you have measured transcripts for respective genes. This is misleading on so many levels and does not reflect the science that you did.

We have rephrased this sentence to be clearer.

C26-L 39: You mention food webs of deep groundwater without any citation for this statement. To the best of my knowledge nobody has ever shown how carbon fluxes work in groundwater and nobody

knows if there actually is a food chain (or if all of the microbiome is just a remnant from sessile microbes?).

We disagree with the reviewer regarding the state of knowledge about the microbiome in groundwater and the existence of food webs in these ecosystems. Although the pace of life in these groundwaters would vary in different ecosystems based on the availability of nutrient and reducing agents, there is accumulated evidence showing that the microbiome of groundwater is active and thriving.

Clues to the existence of food webs are shown by the proof of in-situ activity of viruses in deep continental subsurface and hydraulic fracture fluids in the recent study of Daly et.al.⁴² (Viruses control dominant bacteria colonizing the terrestrial deep biosphere after hydraulic fracturing). Viruses are strong top down control forces in the food web and their in-situ activity is a clear proof that the food web exists in deep continental subsurface including groundwater. There are also other studies that have reconstructed viral genomes from groundwater samples such as Al-Shayeb et.al.⁸ (Clades of huge phages from across Earth's ecosystems). Phages have also been isolated from Äspö HRL^{43,44}. An additional study at the Olkiluoto site studied in this manuscript by Bell et al.⁴⁵ showed biogeochemical cycling of e.g. carbon in the deep groundwater. While the deep groundwater food web might be simpler than typical surface water food web, specifically regarding the presence of nanoflagellates and protists that are more sensitive to the entropic pressure of rock fractures, evidence supports its existence.

C27- L 40: Euks are rather an exception in the deep subsurface. I'd suggest to rephrase unless you show that this is really the case for your system as well.

We wish to point out prokaryotes in this opening sentence as we state that they are at the base of the food web. We also believe it is necessary to mention that eukaryotes¹³ and viruses are also found in these ecosystems as eukaryotic rRNA and mRNA reads have been recovered from groundwater metatranscriptomes¹⁰ and that phages have been isolated from the Äspö HRL^{43,44}.

C28- L 42: The deep groundwater ecosystems are not one environment. These are many. Please rephrase.

We have now rephrased this sentence.

C29- L 45: There is no such thing as one deep core groundwater microbiome. It varies across the entire planet. Only functions are conserved as in almost all ecosystems that are similar.

We have now rephrased this sentence and removed the word "core". We agree with the reviewer that the degree of overlap in microbiome of different groundwaters (i.e. core microbiome) varies for different sites depending on a variety of factors. One additional factor defining the degree of overlap is the resolution of studies. At higher taxonomic levels such as phylum level, these overlaps will be more extensive compared to when high resolution studies at the species level are performed, such as in this study. Here we rely on genome resolved analysis at the level of species and show a core microbiome in these sites containing 165 MAGs/SAGs that were present in groundwater samples at both sites (**Fig. 3 & 1D**). These prevalent MAGs/SAGs, group into 73 genome clusters taxonomically affiliated to both domain Archaea (phyla Nanoarchaeota (class Woesearchaeia) and Thermoplasmata; $n=15$) and Bacteria (phyla Acidobacteriota, Actinobacteriota, Bacteroidota, Caldatribacteriota, Campylobacterota, CG03, Chloroflexota, Desulfobacterota, Firmicutes_A, Nitrospirota, Omnitrophota, Patescibacteria, and Proteobacteria; $n=150$) (See **Supplementary Fig. S2** and **Supplementary TableS4** for details of the common core microbiome). Existence of the same species with almost identical ($\geq 95\%$ ANI over $\geq 70\%$ of the genome length) metabolic context in both locations is a novel and striking finding presented and discussed in our manuscript. In addition, during the revision and exploring public databases we found lineages where all available genomic

representatives exclusively originate from groundwater samples from around the world. One example of these groups is phylum UBA9089 where all available genomes in the GTDB originate from groundwater samples (see more details in C37 below). We believe that this is further support for the hypothesis that the deep biosphere has a core microbiome that extend even beyond sites with similar lithology. However, the degree to which this core microbiome is shared in different locations would vary according to many other concurrent physicochemical and geological factors that are also impacting the diversity of biome.

Considering the last sentence of the reviewer's comment assuming that microbes carry out the functions at least in the biological context we are discussing here, one can argue that seeing functions as conserved features of almost all similar ecosystems is no different from actually seeing a core microbiome in these ecosystems. Functions cannot really be considered as independent of the biological context. It is true that functional redundancy (i.e. different microbes carrying out the same function) is a relevant concept here, but one major consideration is that the rest of the metabolic context between these seemingly functionally redundant groups could be different and impact the realization of these functions in the ecosystem.

A good example of the impact of the background metabolic context on a specific function is seen for the Wood-Ljungdahl pathway. The background metabolic context of the cell defines the direction of this reaction and its role in the cell to drive either autotrophic carbon assimilation or acetate oxidation.

In a direct quote from reference⁴⁶... "The Wood-Ljungdahl pathway is found in a broad range of phylogenetic classes, and is used in both the oxidative and reductive directions. The pathway is used in the reductive direction for energy conservation and autotrophic carbon assimilation in acetogens. When methanogens grow on $H_2 + CO_2$, they use the Wood-Ljungdahl pathway in the reductive direction (like acetogens) for CO_2 fixation; however, they conserve energy by the conversion of $H_2 + CO_2$ to methane. Given that hydrogenotrophic methanogens assimilate CO_2 into acetyl-CoA, it is intriguing that they do not make a mixture of methane and acetate. Presumably this is governed by thermodynamics, since the formation of methane is -36 kJ/mol more favorable than acetate synthesis. Aceticlastic methanogens exploit this advantageous equilibrium to generate metabolic energy by interfacing the Wood-Ljungdahl pathway to the pathway of methanogenesis. In this reverse direction, the combined actions of acetate kinase and phosphotransacetylase catalyze the conversion of acetate into acetyl-CoA. Sulfate reducing bacteria also run the Wood-Ljungdahl pathway in reverse and generate metabolic energy by coupling the endergonic oxidation of acetate to H_2 and CO_2 to the exergonic reduction of sulfate to sulfide ($\Delta G^{0'} = -152$ kJ/mol), with the overall process represented $SO_4^{2-} + CH_3COO^- + 2H^+ \rightleftharpoons HS^- + 2CO_2 + 2H_2O$ $\Delta G^{0'} = -57$ kJ/mol".

Interestingly, we have such an example in our dataset; different representatives affiliated to phyla UBA9089 and Desulfobacterota harbor the genetic potential for dissimilatory sulfate reduction and reductive acetyl-CoA pathway (Wood-Ljungdahl pathway). This metabolic potential enables them to use acetate as a carbon and electron source via reverse acetogenesis. Acetate oxidation to $H_2 + CO_2$ is expected to be thermodynamically unfavorable, but these organisms have the possibility to couple this to sulfate reduction rendering the combined process thermodynamically favorable. This clearly illustrates the importance of looking at functions in the context of whole cell metabolism rather than standalone entities.

In conclusion, we believe that while functions are conserved in almost all similar ecosystems, the role of biological component and their background metabolic context in the realization of these functions should also be acknowledged.

C30- L 47: The authors state again that the deep disconnected biosphere is one environment. This is simply wrong.

Fixed.

C31- L 57/58: Habitats cannot lack a method. Consider rephrasing.

We have now rephrased this sentence.

C32- L 60: please elaborate on how these tunnels might influence the deep biosphere that you sample (e.g., via oxygen).

We have added additional text to the supplementary methods addressing how the two sites are ideally suited for studying the natural, indigenous microbes of the Fennoscandian Shield. The added text reads as below.

Building of the Äspö Hard Rock Laboratory was initiated in 1986 and the site circumvents many problems normally associated with sources of contamination in groundwater. While flow of groundwater towards the tunnel results in mixing of some groundwaters, it also flushes away any anthropogenic contamination introduced into the deep biosphere. In addition, boreholes drilled far into the bedrock means that oxygen does not penetrate to the sampled fracture waters and enable waters to be sampled under *in situ* conditions. Finally, the risk of influencing the microbial community due to materials used to close the borehole sections⁴⁷ is minimized by flushing three section volumes prior to sampling.

Groundwater is accessed via deep drillholes at Olkiluoto fitted with multipackers that allow isolation of fracture fluids. Fractures were pumped for 2–3 months prior to microbiological sampling. During this time, chemical parameters (including dissolved O₂ and oxidation-reduction potential) were monitored to ensure that the water was representative of the isolated fracture. Deep drillholes and excavation of the underground tunnels at Olkiluoto can form transient drawdown of groundwater in connected fractures due to changes in hydraulic head (pressure). This can also cause mixing of different groundwater-types at Olkiluoto, but deep groundwaters are highly reducing and there is no evidence of oxygen penetration⁴⁸. Sterility and anoxic conditions during sampling of groundwaters at Olkiluoto, Finland, were ensured by filtering directly through 0.2 µm filters as described in³².

C33- L 65: Is there one biome in these different sampling sites? That might be a stretch to state in the introduction.

We have now changed it to biomes to highlight that we are not claiming that these two sites have identical composition and distribution of taxa and biomass. We wish to highlight that we are aware of the strength of our methods and are only describing the biological component of these sites.

C34- L 64 – 68: The authors study two sites but draw conclusions for the entire deep biosphere. The manuscript needs to be toned down drastically.

We discuss only according to our results and the sites we studied here. We rephrased this sentence to make this clearer.

C35- L 75: Where are the data on the characteristics of the groundwater? Stating that they have different characteristics without showing them is not scientific. Also add statistical tests and replicates to show the differences. Please compare the characteristics to other study sites to show that your groundwater is representative for the majority of the deep biosphere.

The groundwater characteristics data were presented in the Supplementary Table S1 of the original submission and the supplementary information document as referred to in the methods section. For clarity, we have also now referred to this information in the main text.

C36- L 80: *Were the MAGs curated regarding GC and coverage of the single scaffolds (in, e.g., ANVIO)? The authors should do that as there might be many biases in the MAGs otherwise (see <https://www.nature.com/articles/s41564-018-0171-1>)*

We performed several pilot analyses prior to selecting the best binning approach for our datasets and later carried out manual curation to ensure the quality of the MAGs to be in the best possible state.

To add more details, in our pilot study we checked three prominent binning methods namely CONCOCT, MaxBin2, and MetaBat2 and then inspected the quality of the reconstructed bins from each method. As is mentioned in the paper the reviewer suggests in this comment, the performance of different binning methods to an extent depends on the biome and samples and that is why we checked several methods. We found that MetaBat2 generated the best results for all datasets, based upon total binned bp and total number of good quality (assessed using CheckM) bins generated (for both Bacteria and Archaea). MetaBat2 uses tetra nucleotide frequency, GC content, and coverage into account for binning. Having several datasets from different time points for each borehole allows for generating differential coverage profiles that we leveraged for binning.

After the in-depth inspection of generated MAGs from these methods, we decided to go with MetaBat2 at the standard quality thresholds of <5% contamination and >50% completeness. The results from this comparison, and the Jupyter notebook used to investigate them, can be accessed at https://github.com/johnne/deepbiosphere_binning_eval.

These bins were further manually checked for the taxonomy of binned 16S rRNA and the taxonomy assigned using GTDB-tk based on marker genes. In cases containing inconsistencies, we removed the whole contig containing 16SrRNA from that bin and rechecked the consistency of GC%, tetra nucleotide frequency, and coverage in all contigs of this bin. We did this manual check since in many cases the 16S rRNA containing contigs tend to get miss-binned.

We also ran RefineM (<https://github.com/dparks1134/RefineM>) developed by Parks et al. to ensure the quality of our bins and consistency of GC and tetra nucleotide distribution in all contigs of each bin.

To be more stringent, we avoided any effort for recruiting contigs to low quality bins to artificially increase the completeness. We preferred to discard low quality bins from the downstream analyses altogether and only include the good quality and manually curated bins for the subsequent analyses. Following this stringent quality-aware strategy and based on the lower quality of bins generated via MaxBin2 and CONCOCT for our samples, we avoid using DAS tool for combining generated bins.

C37- L 93: please compare your diversity to the diversity that other studies found. Is your data representative of the deep biosphere?

As we have mentioned in the manuscript, the diversity (both species richness and evenness) in deep groundwaters will depend on many factors such as bedrock lithology, available electron donors and acceptors, depth, and hydrological isolation from the photosynthesis-fueled surface (line 41-42 of the text and thereafter in many other places). One major point of this paper is to compare deep

disconnected groundwater running through similar lithologies to inspect their diversity. Our samples coming from disconnected locations in the same bedrock with independent sample collection and processing allow to identify shared core populations in oligotrophic Fennoscandian Shield fracture waters.

Other existing groundwater datasets shown in the Supplementary Figure S1 originate from different geologies and belong to shallow aquifers, CO₂ saturated geysers, mines, and carbon-rich shales. The biota in such ecosystems will have very different diversity. Consequently, this will make it rather difficult to find a representative for the deep groundwater diversity to compare all other datasets to. In this study, we show that there is an overlap between the diversity of our two sites located in the same lithology and that these deep groundwaters offer fixed niches that can select for similar microbes.

In addition, to answer this question we have explored all publicly available deep groundwater genomes in more detail by searching public databases to recover groundwater microbes and to compare them to our reconstructed FSGD MAGs/SAGs. In doing so we found lineages where all available genomic representatives exclusively originate from groundwater samples from around the world. One example of these groups is phylum UBA9089 where all available genomes in the GTDB originate from groundwater samples (see the table below). Interestingly, we recovered 14 MAGs and 1 SAG belonging to this taxon in our FSGD dataset and found this taxon to be one of the highly transcribing taxa in our datasets according to metatranscriptome data. We believe that this is further support for the hypothesis that the deep biosphere has a core microbiome that extends beyond sites with similar lithology. However, the degree to which this core microbiome is shared in different locations would vary according to many other concurrent physicochemical and geological factors that are also impacting the diversity of biomes.

We have now added this section to the supplementary documents and the manuscript text. We are aware that resolving the degree of overlap in the whole groundwater core microbiome would need more in depth and concerted analysis of all available groundwater samples and believe the reviewer will agree that this is beyond the scope of the present manuscript.

ID	GTDB Taxonomy	Biosample	origin
GCA_001873945.1	d_Bacteria; p_UBA9089; c_CG2-30-40-21; o_CG2-30-40-21; f_CG2-30-40-21; g_CG2-30-40-21; s_CG2-30-40-21 sp001873945	SAMN04328251	USA: Crystal Geyser near Green River, Utah
GCA_002772295.1	d_Bacteria; p_UBA9089; c_CG2-30-40-21; o_CG2-30-40-21; f_CG2-30-40-21; g_CG2-30-40-21; s_CG2-30-40-21 sp001873945	SAMN06659668	USA: Crystal Geyser near Green River, Utah
GCA_002783685.1	d_Bacteria; p_UBA9089; c_CG2-30-40-21; o_CG2-30-40-21; f_CG2-30-40-21; g_CG2-30-40-21; s_CG2-30-40-21 sp001873945	SAMN06659921	USA: Crystal Geyser near Green River, Utah
GCA_002784185.1	d_Bacteria; p_UBA9089; c_CG2-30-40-21; o_CG2-30-40-21; f_CG2-30-40-21; g_CG2-30-40-21; s_CG2-30-40-21 sp001873945	SAMN06659863	USA: Crystal Geyser near Green River, Utah
GCA_002790015.1	d_Bacteria; p_UBA9089; c_CG2-30-40-21; o_CG2-30-40-21; f_CG2-30-40-21;	SAMN06660059	USA: Crystal Geyser near

	g_CG2-30-40-21; s_CG2-30-40-21 sp001873945		Green River, Utah
GCA_003485015.1	d_Bacteria; p_UBA9089; c_UBA9089; o_UBA9089; f_UBA9089; g_UBA9089; s_UBA9089 sp003485015	SAMN08018309	Filtered fracture water from deep in South Africa depth 1900m
GCA_003498085.1	d_Bacteria; p_UBA9089; c_UBA9088; o_UBA9088; f_UBA9088; g_UBA9088; s_UBA9088 sp003498085	SAMN08018308	Filtered fracture water from deep in South Africa depth 1900m
GCA_005239945.1	d_Bacteria; p_UBA9089; c_CG2-30-40-21; o_CG2-30-40-21; f_SBAY01; g_SBAY01; s_SBAY01 sp005239945	SAMN08774410	Groundwater from mouth of Comano Thermal Spring, Italy, Trento

C38- L 104: It is problematic to map reads against all created population genomes as there might be redundancy. The authors should de-replicate their genome dataset based on ANI.

We have clustered our genome dataset based on ANI at $\geq 95\%$ threshold over $\geq 70\%$ coverage of the genome length. The mapping results are clumped together for each cluster taking into account the covered length of genomes (we have updated the mapping calculation according to your suggestion and for more details, please check the response to comment C40). As mentioned in the text, we only use presence/absence rather than abundance values making de-replication an unnecessary step.

C39- L 119: The authors suddenly talk about size fractionation. Please explain the rationale behind the study setup and how this fractionation impacted your sample normalization strategy (which I assume has been done based on number of sequenced base-pairs?).

To clarify this point, we have added a short description to the methods section and refer to the detailed description of samples and sample processing methods that are presented in the supplementary information document and Supplementary Table S1 in the methods section of the manuscript.

Regarding the normalization, the reads recruited to each genome were normalized as TPM (Transcripts Per Kilobase Million) and this normalization helps with bypassing the issues the reviewer is raising. TPM is similar to RPKM (Reads Per Kilobase Million) and FPKM (Fragments Per Kilobase Million) and the only difference is the order of operations. To calculate TPM we follow these steps:

1-Divide the read counts by the length of each contig in kilobases. This gives reads per kilobase (RPK). 2-Sum all the RPK values in a sample and divide it by 1,000,000. This is “per million” scaling factor. 3-Divide the RPK values by the “per million” scaling factor. This gives TPM.

When calculating TPM, the only difference is that we normalize for contig length first, and then normalize for sequencing depth. However, the effects of this difference are quite profound. When using TPM, the sum of all TPMs in each sample are the same. This makes it easier to compare the

proportion of reads that mapped to a genome in each sample. In contrast, with RPKM and FPKM, the sum of the normalized reads in each sample may be different, and this makes it harder to compare samples directly. In addition to this, we only use presence/absence patterns for our analysis, which is a much more robust approach, particularly when the covered length of each genome is taken into account in the new mapping.

C40- L 121: The authors determine the presence/absence of a genome based on mapping coverage. However, mapping coverage is meaningless here. The authors should calculate how much of the genome is actually covered by reads rather than how many reads have mapped (many reads can map in one region leaving other regions uncovered).

As per reviewer's suggestion, we have now repeated the mapping step and taken the fraction of genome covered by the mapped reads into account when defining the presence/absence patterns of our reconstructed MAGs/SAGs. In this new approach, we only consider a contig present if $\geq 50\%$ of the contig length is covered by the mapped reads. We have updated the text according to the results of these highly stringent thresholds for defining the presence/absence patterns and identify a common core microbiome. Figures, supplementary table, and the text have been updated according to the new results.

We wish to highlight here that these presence/absence patterns of the core microbiome are also supported by the assembly of representatives of the same genome cluster (species) from both locations (15 clusters) (Supplementary Table S4).

C41- L 132: You make a general statement here but you only show this for two sites. Please rephrase / tone down.

We have changed this to emphasis again that we are discussing the two sites studied here.

C42- L 135: If something doesn't confute a possibility it cannot be used as evidence but that's how you argue. Please do provide solid evidence for evolutionary convergence between the sites.

Our conclusion based on the results of our analyses is that the convergence we see in the core microbiome of the Fennoscandian Shield deep groundwater community is rather ecological. One reason for this can be the relatively high phylogenetic diversity of the core microbiome we recovered in our analysis. However, to be cautious and not confute the possibility of an evolutionary convergence due to the long residence time in these ecosystems, we also checked whether we could find clues toward evolutionary adaptations in the microbial community of these ecosystems. Indeed, we detect some signal of evolutionary adaptations (e.g., responding to the increasing salinity by changing isoelectric points of proteome) detectable in our genomes and therefore, cannot be so categorical as to reject the possibility for evolutionary species convergence. Determining the extent of these evolutionary adaptations and their role in the fitness needs more in depth in-situ experiments. We have now rephrased this to avoid confusion.

Line 133-137- The relatively high phylogenetic diversity of the prevalent species implies a significant role of ecological convergence (due to e.g., availability of nutrient and reducing agents) rather than evolutionary responses. However, this by no means confutes the possibility of an evolutionary convergence as the community clearly undergoes adaptation over the long residence time characteristic for deep groundwaters.

C43- L 135: Please demonstrate that the two sites were not connected in the past (elaborate on geology / geological history of the sites).

We have added a section to better describe these sites to the supplementary information document under the headline "**Site lithologies**" that also provides the geological history of these sites.

C44-L137-139: Statistics missing.

In this section, we report the salinities measured in our samples (also shown in the Supplementary Table S1) ranging between 0.4 to 1.8%. Just to put these values in the context of aquatic salinity we also mention the salinity of Baltic proper (0.5%) and marine water (3%). These are the direct measurements reported in this section and as it is shown in the Supplementary Table S1 column for salinity (ppt) they remain stable for each borehole

C45-L 141: Please elaborate how you determine that these are evolutionary timescales.

We have changed this text to highlight the long residence time of our water samples.

C46-L 142: You claim that basic proteins are enriched in one sample. Please provide statistical tests for this.

Kolmogorov–Smirnov test results show a significantly different distributions for all these samples ($p < 10^{-26}$, for all comparisons). We mention that the frequency of basic proteins in the sample OL-KR46 decreases compared to the other samples from this location and this could be potentially related to the increase in the salinity in this sample. This response to increased salinity is also shown in work by Cabello-Yeves and Rodriguez-Valera when they show changes in the proteome in response to marine-freshwater transition⁴⁹.

The isoelectric point trend towards a reduced prevalence of basic proteins with increasing salinity is specifically pronounced in the Olkiluoto drillhole OL-KR46 where salinity reaches a maximum of 1.8% (Fig. 4B). This is reflected in the text (lines 161-163)

C47-L 145: You claim the two samples are similar. Provide a statistical test (e.g., Power test).

We have now performed Kolmogorov–Smirnov test to inspect these distributions and the significance of their differences. Our results show that all these distributions are significantly different ($p < 10^{-26}$, for all comparisons). The pairwise Kolmogorov's D value show a higher distribution similarity between KR11 and KR13 compared to KR46. Also, it shows higher distribution similarity between MM, OS, and UM themselves. The table presented here also shows the pairwise values. We also included the raw data for isoelectric points in the supplementary material as data source.

	KR11	KR13	KR46	MM	OS	UM
KR11.D	0	0.018273	0.105767	0.065185	0.047085	0.077652
KR13.D	0.018273	0	0.115343	0.083054	0.058814	0.093379
KR46.D	0.105767	0.115343	0	0.04998	0.059179	0.038508
MM.D	0.065185	0.083054	0.04998	0	0.037869	0.020039
OS.D	0.047085	0.058814	0.059179	0.037869	0	0.039958
UM.D	0.077652	0.093379	0.038508	0.020039	0.039958	0

C48-L 148: Species richness cannot be determined from binned genomes. Maybe that one sample with low richness was problematic in good binning? Please add an analysis of species richness of all samples based on marker genes. Do not use ribosomal RNA for that as it doesn't get assembled well. Use a house keeping gene that is good for species delineation, e.g. *gyrB*

In this context, we are discussing the richness derived from a presence/absence matrix that was generated by including all high quality MAGs ($n=1392$) reconstructed from all metagenomes and

calculated their presence/absence in all metagenomes. We also want to show the status of data that is being compared in this context rather than the richness of ecosystem *per se*.

We agree with the reviewer that the binned genomes are not the best representation of the species richness in an ecosystem and have performed the suggested analysis using the assembled *gyrB* genes. We extracted the *gyrB* genes from all metagenomic assemblies, evaluated the annotation by confirming the conserved domains of the gene, and then clustered them at the 97% and 88% identity threshold defined for this gene to reconstruct *gyrB* mOTUs^{50,51}. The same plot derived from the *gyrB* richness analysis showed a similar overall trend as the genome clusters (Fig.1)

The methods and plots of this analysis have been added to the supplementary information document under the headline “Species richness using gene *gyrB*”

Figure1- Species richness in metagenomes generated from each ecosystem based on gene *gyrB*.

C49-L 161: “these groundwaters” -- do these references refer to the sites that you sampled here as well? If so, please perform a comparison of the respective findings! If not, please rephrase.

We have now rephrased this sentence.

C50-L 169: Sentence ending with citations 22/23. Please elaborate more on why this is the case otherwise the reader won't understand your message.

Thanks for your suggestion. We have now expanded this section to make our point clearer.

C51-L 188: Not only viruses! CRISPR-Cas systems can work against all mobile genetic elements. We have now fixed this sentence to also mention this.

C52-L 190: use past tense “expressed”

Fixed.

C53- L 194-196: pure speculation, no evidence for this.

We believe that this section is a data informed conclusion based on the transcribed profile of this genome. The expressed genes show a codon usage bias and annotate as CRISPR/Cas system-associated proteins Cas8a1 and Cas7 that could be activated in response to phage or other types of foreign DNA. In addition, we also see ribonuclease toxin, BrnT of type II toxin-antitoxin system in the transcribed profile of this genome. This toxin is known to respond to environmental stressors and cease bacterial growth by rapid attenuation of the protein synthesis. Based on this expression profile (the detailed annotation of the expressed genes of this genome is shown in the Supplementary Table S5), we argue that this cell could be exposed to phage infection.

C54- L 216: Using the term Nanoarchaeota from GTDB is extremely misleading. I suggest using a term that does not have a historic connection to a specific organism group, like Woesearchaeota etc.

We opted for using GTDB taxonomy throughout the text to be consistent. However, as per reviewer's suggestion, we have now added class level info for Archaea when we refer to them in the text.

C55- L 219: Please provide evidence for leaky metabolites (metabolomics) and elaborate more or remove the statement. This has not been shown before and is a huge claim.

We have rephrased this sentence based on the reviewer's suggestion. At this time we do not have metabolite analyses from these sites but are working on this in follow up projects.

C56- L 223: Lipid and nucleotide metabolism is usually sparse in Patescibacteria. Are these binning errors since you didn't manually check all the MAGs? Please elaborate more.

For discussing these annotations, we have used KEGG modules. For example, the Lipid metabolism module contains all these sub modules (00061 Fatty acid biosynthesis, 00062 Fatty acid elongation, 00071 Fatty acid degradation, 00072 Synthesis and degradation of ketone bodies, 00073 Cutin, suberine and wax biosynthesis, 00100 Steroid biosynthesis, 00120 Primary bile acid biosynthesis, 00121 Secondary bile acid biosynthesis, 00140 Steroid hormone biosynthesis, 00561 Glycerolipid metabolism, 00564 Glycerophospholipid metabolism, 00565 Ether lipid metabolism, 00600 Sphingolipid metabolism, 00590 Arachidonic acid metabolism, 00591 Linoleic acid metabolism, 00592 alpha-Linolenic acid metabolism, 01040 Biosynthesis of unsaturated fatty acids). While our MAGs/SAGs do not encode any complete biosynthesis pathways for fatty acid based lipid biosynthesis or nucleotide biosynthesis, they do express some genes that fall into the lipid metabolism module according to KEGG module organization. We see how this can cause confusion and thank the reviewer for pointing this out. We have now rephrased this carefully throughout the text to avoid confusion.

C57- L 236: I do not agree with the conclusion that your data underscores cooperation as a competent evolutionary strategy. The easiest answer is, that the episybionts can have multiple hosts or do not rely on hosts at all and cooperation is not relevant, rather cell death.

We agree with the reviewer that necromass plays a partial role in sustaining life in the deep oligotrophic groundwater. However, with the long residence time of water in these deep groundwaters, necromass cannot be the only support for sustaining life over time as there is limited biomass to turn into necromass and the old saline waters in particular have very limited connection to the photosphere for bringing in new biomass. Additionally, the necromass would not be exclusively available to symbionts and its presence will initiate competition between resident microbes.

On the other hand, having multiple hosts is not against symbionts adopting a cooperation strategy. We do not mention an exclusive partnership for these symbionts but emphasize that they can be promiscuous and have different partners in different ecosystems or multiple partners in one ecosystem and in both cases, they benefit from cooperation with their hosts/partners. Having these

putative symbionts as a part of core microbiome in the two sites we studied here hint at a very well adapted/competitive lifestyle in these groups.

The reviewer also validly argues that Patescibacteria cells might not be episympiotic or cooperate with other members of the community. To build on the existing knowledge about these groups we have now checked for available clues to inspect this in our FSGD database of MAGs/SAGs. While we find genes related to lipid metabolism according to the KEGG module organization we could not find any MAG/SAG related to CPR taxa in our FSGD database that encodes genomic potential for a complete fatty acid (FA)-based lipid biosynthesis pathway yet still these cells have fatty acids in their membrane. This has been discussed in a study by Probst & Elling et al.³⁶ where they also mention “Other putative bacterial and archaeal symbionts from different branches of the tree of life also do not encode for their own lipid biosynthesis pathway and at least one hyperthermophilic episympiotic (*Nanoarchaeum equitans*) has been suggested to acquire its lipids from the host archaeon. However, the origin and types of lipids used by CPR bacteria remain elusive.”

Probst & Elling et. al. in their work suggest that these lipids may originate from other community members. This is one important cooperation between CPR representatives and other community members that could provide them with this necessary cellular component.

Most cultures and electron micrographs available from representatives of CPR bacteria and DPANN archaea provide more support for them having an symbiotic lifestyle as it is reviewed by Castelle & Banfield as an unusual lifestyle that is predicted to be common for CPR and DPANN taxa³⁵. Considering the state of our knowledge and what metabolic capabilities we find in our reconstructed MAGs/SAGs we can suggest a symbiotic lifestyle for our MAGs/SAGs affiliated to these taxa.

We have also added some of this discussion to the text in this section.

C58-L 281: Since these genes build the core of your manuscript, please elaborate how many genomes of the community possess these genes! Add statistics.

We have now added the stats of this gene in the text. Additionally, we have now added the list of MAGs/SAGs that we recovered this gene from as Supplementary Table S12.

C59-L 281: Please perform a phylogeny as well as a motif search for the active center of the recovered proteins to demonstrate that they are conserved and are likely performing the predicted function. Modelling of the protein 3D structure could demonstrate the robustness of your results.

The phylogeny of the DnaE2 gene is presented in Supplementary Figure S7. We have now also checked the conserved domains and structure of the DnaE2 proteins and all annotated genes were approved for putative DnaE2 annotation

C60-L 284: Period missing.

Fixed.

C61- The authors should make all genomic and transcriptomic data publicly available.

All the data are publicly available and archived in figshare together with all the phylogeny alignments and tree files in one project for easy access and batch download. Also, all metagenomes and metatranscriptomes and reconstructed MAGs and SAGs are submitted to GenBank and the accession numbers presented in the Supplementary Table S1.

Reviewer #3

C62- 1. A fundamental finding is presented in the form of the core microbiome for deep oligotrophic groundwaters. But this is never defined in the study. Which members constitute the core microbiome and what is the extent of overlap of their niches? Supplementary Table 4 contains a lot of raw data, but the authors should clearly define the “core” microbiome instead of letting the reader interpret this.

We have now added the information about the common core microbiome to the results and included Supplementary Figure S2 to show the degree of overlap as a Venn diagram as well as their taxonomy. We have also decluttered the Supplementary TableS4.

Supplementary Figure S2- Overlap of the microbiome the Fennoscandian Shield deep groundwater flowing in two disconnected sites.

C63- 2. A second fundamental finding is presented in the form of “description of ecological and evolutionary processes essential for successfully occupying and propagating in the available niches.”. Again, I found this underwhelming. While a number of the findings such as the potential regulation of transcription are intriguing – there is currently insufficient evidence presented. Some comparisons and potentially answering questions such as “Are the number of Toxin/Anti toxin systems greater than one would typically observe in other environments like shallow aquifers?” would bolster this argument.

As per reviewer’s suggestion, we have now collected 2402 metagenome-assembled genomes generated by Anantharaman et al. from samples collected from an aquifer adjacent to the Colorado River near Rifle, CO, USA, at the Rifle Integrated Field Research site¹. We have annotated these MAGs and screened them for toxin/antitoxin system genes. We then compared the distribution of these toxin/antitoxin systems to our deep oligotrophic groundwater.

The prevalence of genes coding for the toxin-antitoxin systems in FSGD MAGs/SAGs compared to shallow aquifer MAGs shows that in majority of cases (20 out of 23) a relatively higher percentage of FSGD MSGs/SAGs encode for genes involved in these systems compared to the Rifle aquifer MAGs (stats shown in **Supplementary Table S11**).

C64- 3. Lines 194-196 It is unclear to be as to why this would be associated with phages defense. Could the BrnT/BrnA toxin-antitoxin system simply have been active in these cells at the same time?

In this section we explore the expressed profile of the SAG 3300021839. Our results show a codon usage bias between the expressed profile of this SAG compared to the rest of its protein coding sequences. These expressed genes annotate as CRISPR/Cas system-associated proteins Cas8a1 and Cas7 that could be activated in response to phage or other types of foreign DNA. In the transcribed profile of this SAG we also see ribonuclease toxin, BrnT of type II toxin-antitoxin system. This toxin is known to respond to environmental stressors and cease bacterial growth by rapid attenuation of the

protein synthesis. Based on this expression profile (the detailed annotations of the expressed genes of this genome are shown in the Supplementary Table S5), we reason that this cell could be exposed to phage infection.

While we only have the BrnT in the expressed profile there are publications showing that both BrnT/BrnA could be co-expressed in *Brucella abortus*⁵².

C65- 4. In a highly energy-limited environment, it is understandable that there may be significant metabolic interactions between organisms, specifically Patescibacteria and Nanoarchaeota. However, some analyses on specific associations (even predicted) of these organisms with their hosts would constitute a significant advance.

We agree with the reviewer that exploring the mode of association for *Patescibacteria and Nanoarchaeota* is very interesting. While there are many studies where co-occurrence networks are used to define the potential partners of these taxa, we believe that this co-occurrence network analysis bear minimal ecological insight in this case. Specifically, since we still do not know the host/partner range for these taxa. Interestingly, in a recent publication entitled “Co-occurrence is not evidence of ecological interactions” by Blanchet et. al. they discuss this in more detail and conclude that “significant spatial associations between species (or lack thereof) is a poor proxy for ecological interactions”⁵³. We try to explain some cases of interactions via metabolic potentials; however, in these cases we cannot prove a direct exclusive metabolic interaction. Consequently, these can only be concluded as a promiscuous interaction.

One thread we tried to follow to respond to this comment was to investigate our generated SAGs. As we show in our manuscript, we generated 564 SAGs from which 114 SAGs fall in the quality range to be included in this manuscript. We explored the rest of these SAGs to see whether we can find cases of *Patescibacteria and Nanoarchaeota* together with other groups. We were more positive about this method as we expected if these epibiont/partner have surface connection they could get sorted together. However, we did not find any promising results for this analysis yet.

C66- 5. Metabolic analyses are not comprehensive. Sulfur species (specifically sulfate) is presented as an important electron acceptor in these systems. Yet, no comprehensive analyses were conducted to verify the metabolic potential for sulfate reduction. Given the reversible nature of pathways associated in sulfate reduction, it is important to conduct these analyses to verify the claims in the study.

Metabolic potential of MAGs/SAGs belonging to the highly transcribing genome clusters are detailed in Figure 5. To expand the description of the sulfur metabolism in more detail we have now added the phylogeny of the *dsrA* gene in order to further evaluate the reductive/oxidative direction of the process. This analysis agreed with our prior analysis for metabolic annotations and has been added to the manuscript as Supplementary Figure S5 the reconstructed tree file is also uploaded to figshare and is publicly available.

Supplementary Figure S5- Reconstructed phylogeny of the DsrA protein sequences recovered from FSGD reconstructed MAGs and SAGs (dark blue) together with reference DsrA sequences (in red). Different types of DsrA protein according to their function are mentioned in the figure.

We further evaluated these annotations by inspecting the active sites and reconstructing phylogeny. We have now added a detailed representation of potential of the highly expressing MAGs/SAGs for dissimilatory sulfur metabolism with including Supplementary Figure S6 for gene presence info in addition to module completeness information of Figure 5.

Module	Entry	Name	C19	C21	C24	C32	C34	C37	C37	C70	C12	C35	C30	C340	C33	C38	C39	C40
M00596	K0095	sat	sulfate adenylyltransferase [EC:2.7.7.4]															
	K0039	aprA	adenylylsulfate reductase, subunit A [EC:1.8.99.2]															
	K0039	aprB	adenylylsulfate reductase, subunit B [EC:1.8.99.2]															
	K1118	dsrA	dissimilatory sulfite reductase alpha subunit [EC:1.8.99.5]															
	K1118	dsrB	dissimilatory sulfite reductase beta subunit [EC:1.8.99.5]															
M00595	K1722	soxA	L-cysteine S-thiosulfotransferase [EC:2.8.5.2]															
	K1722	soxX	L-cysteine S-thiosulfotransferase [EC:2.8.5.2]															
	K1722	soxB	S-sulfosulfanyl-L-cysteine sulfohydrolase [EC:3.1.6.20]															
	K1722	soxC	sulfane dehydrogenase subunit SoxC															
	K2262	soxD	S-disulfanyl-L-cysteine oxidoreductase SoxD [EC:1.8.2.6]															
	K1722	soxY	sulfur-oxidizing protein SoxY															
	K1722	soxZ	sulfur-oxidizing protein SoxZ															

Supplementary Figure S6- Detailed annotation of highly expressing genome clusters for dissimilatory sulfate reduction (M00596) and thiosulfate oxidation by SOX complex (M00595) modules. The DsrA gene annotations were further confirmed by reconstructing phylogeny of the annotated genes (Supplementary Figure S5) and only the presence of reductive DsrA is shown here.

C67- 6. *This study seems to be presented in a vacuum. It seems that the authors have recently published similar papers on sulfur cycling and metatranscriptomes from these environments. Some background demonstrating the advances made by this current study and contrasts with the previous study will provide some useful context.*

These sites and their biological component have been separately studied by the coauthors for several years. In this manuscript, we are presenting an extensive comparative study of two sites in the Fennoscandian Shield. Apart from the novelty of performing a comparative study of deep oligotrophic groundwater in two deep sites in one bedrock we have incorporated (i) an extensive multi-omics study; (ii) we have developed a genomic database (FSGD) and studied their distribution and convergence in these two sites that is a new frontier in subsurface studies specifically sub-terrestrial ecosystems; and () instead of annotating functions to short metatranscriptome reads in this study we map the metatranscriptome reads against our FSGD MAGs and SAGs and study highly transcribed genes and their function in the context of overall cell metabolism (this method is rather novel and not widely applied to aquatic ecosystems specifically groundwater). This enables us to refine and evaluate the overall niches that allow for high expression in some microbes. This information gives valuable clues toward the nature of deep groundwater niches at large and those occupied by the core microbiome specifically. This extensive study allows for more in depth understanding of these ecosystems and allows for developing hypotheses on the microbe's mode of life in these ecosystems.

C68- 7. *Figure 6 is not present in the manuscript. I have gone over the individual files as well and cannot find this.*

We apologize for this oversight that has now been fixed.

C69- 8. *Line 336 Is the coverage threshold accurate?*

This coverage is a part of the threshold used for clustering reconstructed MAGs and SAGs in line with the operational definition of the species. For this purpose we have used Average Nucleotide Identity at $\geq 95\%$ and the coverage of $\geq 70\%$.

References:

1. Anantharaman, K. *et al.* Thousands of microbial genomes shed light on interconnected biogeochemical processes in an aquifer system. *Nat. Commun.* **7**, 1–11 (2016).
2. Castelle, C. J. *et al.* Extraordinary phylogenetic diversity and metabolic versatility in aquifer sediment. *Nat. Commun.* **4**, 2120 (2013).
3. Probst, A. J. *et al.* Differential depth distribution of microbial function and putative symbionts through sediment-hosted aquifers in the deep terrestrial subsurface. *Nat. Microbiol.* **3**, 328– 336 (2018).
4. Daly, R. A. *et al.* Microbial metabolisms in a 2.5-km-deep ecosystem created by hydraulic fracturing in shales. *Nat. Microbiol.* **1**, 16146 (2016).
5. Sackett, J. D. *et al.* Four draft Single-Cell genome sequences of novel , nearly identical Kiritimatiellaota strains isolated from the continental deep subsurface. *Microbiol Resour*

Announc e01249-18 (2019).

6. Flemming, H.-C. & Wuertz, S. Bacteria and archaea on Earth and their abundance in biofilms. *Nat. Rev. Microbiol.* **17**, 247–260 (2019).
7. Magnabosco, C. *et al.* A metagenomic window into carbon metabolism at 3 km depth in Precambrian continental crust. *ISME J.* **10**, 730–741 (2016).
8. Al-Shayeb, B. *et al.* Clades of huge phages from across Earth’s ecosystems. *Nature* (2020) doi:10.1038/s41586-020-2007-4.
9. Schwank, K. *et al.* An archaeal symbiont-host association from the deep terrestrial subsurface. *ISME J.* **13**, 2135–2139 (2019).
10. Lopez-Fernandez, M. *et al.* Metatranscriptomes reveal that all three domains of life are active but are dominated by Bacteria in the Fennoscandian crystalline granitic continental deep biosphere. *MBio* **9**, 1–15 (2018).
11. Wu, X. *et al.* Potential for hydrogen-oxidizing chemolithoautotrophic and diazotrophic populations to initiate biofilm formation in oligotrophic, deep terrestrial subsurface waters. *Microbiome* **5**, 1–13 (2017).
12. Engelhardt, T., Kallmeyer, J., Cypionka, H. & Engelen, B. High virus-to-cell ratios indicate ongoing production of viruses in deep subsurface sediments. *ISME J.* **8**, 1503–1509 (2014).
13. Borgonie, G. *et al.* Eukaryotic opportunists dominate the deep-subsurface biosphere in South Africa. *Nat. Commun.* **6**, (2015).
14. Jørgensen, B. B. Shrinking majority of the deep biosphere. *Proc. Natl. Acad. Sci. U. S. A.* **109**, 15976–15977 (2012).
15. Li, J. *et al.* Recycling and metabolic flexibility dictate life in the lower oceanic crust. *Nature* **579**, 250–255 (2020).
16. Heard, A. W. *et al.* South African crustal fracture fluids preserve paleometeoric water signatures for up to tens of millions of years. *Chem. Geol.* **493**, 379–395 (2018).
17. Robador, A. *et al.* Nanocalorimetric characterization of microbial activity in deep subsurface oceanic crustal fluids. *Front. Microbiol.* **7**, (2016).
18. Bradley, J. A., Amend, J. P. & LaRowe, D. E. Survival of the fewest: Microbial dormancy and maintenance in marine sediments through deep time. *Geobiology* **17**, 43–59 (2019).
19. Wasmund, K., Mußmann, M. & Loy, A. The life sulfuric: microbial ecology of sulfur cycling in marine sediments. *Environ. Microbiol. Rep.* **9**, 323–344 (2017).
20. Momper, L. *et al.* Major phylum-level differences between porefluid and host rock bacterial communities in the terrestrial deep subsurface. *Environ. Microbiol. Rep.* **9**, 501–511 (2017).
21. Suko, T. *et al.* Geomicrobiological properties of Tertiary sedimentary rocks from the deep terrestrial subsurface. *Phys. Chem. Earth* **58–60**, 28–33 (2013).
22. Arbour, T. J., Gilbert, B. & Banfield, J. F. Diverse Microorganisms in Sediment and Groundwater Are Implicated in Extracellular Redox Processes Based on Genomic Analysis of Bioanode Communities. *Front. Microbiol.* **11**, (2020).
23. Borgonie, G. *et al.* New ecosystems in the deep subsurface follow the flow of water driven by geological activity. *Sci. Rep.* **9**, 1–16 (2019).
24. Lopez-Fernandez, M., Åström, M., Bertilsson, S. & Dopson, M. Depth and Dissolved Organic Carbon Shape Microbial Communities in Surface Influenced but Not Ancient Saline Terrestrial Aquifers. *Front. Microbiol.* **9**, 1–16 (2018).

25. Cai, L. *et al.* Active and diverse viruses persist in the deep sub-seafloor sediments over thousands of years. *ISME J.* **13**, 1857–1864 (2019).
26. Lopez-Fernandez, M., Broman, E., Simone, D., Bertilsson, S. & Dopson, M. Statistical Analysis of Community RNA Transcripts between Organic Carbon and Geogas-Fed Continental Deep Biosphere Groundwaters. *MBio* **10**, e01470-19 (2019).
27. Lopez-Fernandez, M. *et al.* Investigation of viable taxa in the deep terrestrial biosphere suggests high rates of nutrient recycling. *FEMS Microbiol. Ecol.* **94**, 1–9 (2018).
28. Bagnoud, A. *et al.* Reconstructing a hydrogen-driven microbial metabolic network in Opalinus Clay rock. *Nat. Commun.* **7**, 1–10 (2016).
29. Lau, M. C. Y. *et al.* An oligotrophic deep-subsurface community dependent on syntrophy is dominated by sulfur-driven autotrophic denitrifiers. *Proc. Natl. Acad. Sci. U. S. A.* **113**, E7927– E7936 (2016).
30. Lever, M. A. *et al.* Life under extreme energy limitation: A synthesis of laboratory- and field-based investigations. *FEMS Microbiol. Rev.* **39**, 688–728 (2015).
31. Wu, X. *et al.* Microbial metagenomes from three aquifers in the Fennoscandian shield terrestrial deep biosphere reveal metabolic partitioning among populations. *ISME J.* **10**, 1192– 1203 (2016).
32. Bell, E. *et al.* Active sulfur cycling in the terrestrial deep subsurface. *ISME J.* **14**, 1260–1272 (2020).
33. Balvoc, M. Comprehensive analysis of DNA polymerase III a subunits and their homologs in bacterial genomes. **42**, 1393–1413 (2014).
34. Castelle, C. J. *et al.* Biosynthetic capacity, metabolic variety and unusual biology in the CPR and DPANN radiations. *Nat. Rev. Microbiol.* **16**, 629–645 (2018).
35. Castelle, C. J. & Banfield, J. F. Major New Microbial Groups Expand Diversity and Alter our Understanding of the Tree of Life. *Cell* **172**, 1181–1197 (2018).
36. Probst, A. J. *et al.* Lipid analysis of CO₂-rich subsurface aquifers suggests an autotrophy-based deep biosphere with lysolipids enriched in CPR bacteria. 1547–1560 (2020) doi:10.1038/s41396-020-0624-4.
37. Huber, H. *et al.* A new phylum of Archaea represented by a nanosized hyperthermophilic symbiont. *Nature* **417**, 63–67 (2002).
38. Wurch, L. *et al.* Genomics-informed isolation and characterization of a symbiotic Nanoarchaeota system from a terrestrial geothermal environment. *Nat. Commun.* **7**, 12115 (2016).
39. Golyshina, O. V. *et al.* ‘ARMAN’ archaea depend on association with euryarchaeal host in culture and in situ. *Nat. Commun.* **8**, 60 (2017).
40. He, X. *et al.* Cultivation of a human-associated TM7 phylotype reveals a reduced genome and epibiotic parasitic lifestyle. *Proc. Natl. Acad. Sci. U. S. A.* **112**, 244–249 (2015).
41. Moreira, D., Zivanovic, Y., López-archilla, A. I. & Iniesto, M. Reductive evolution and unique infection and feeding mode in the CPR predatory bacterium *Vampirococcus lugosii*. *Bioarxiv* (2020) doi:<https://doi.org/10.1101/2020.11.10.374967>.
42. Daly, R. *et al.* Viruses control dominant bacteria colonizing the terrestrial deep biosphere after hydraulic fracturing. *Nat. Microbiol.* **4**, 352–361 (2019).
43. Kyle, J. E., Eydal, H. S. C., Ferris, F. G. & Pedersen, K. Viruses in granitic groundwater from 69 to 450 m depth of the spö hard rock laboratory, Sweden. *ISME J.* **2**, 571–574 (2008).

44. Eydal, H. S. C., Jägevall, S., Hermansson, M. & Pedersen, K. Bacteriophage lytic to *Desulfovibrio aespoeensis* isolated from deep groundwater. *ISME J.* **3**, 1139–1147 (2009).
45. Bell, E. *et al.* Biogeochemical cycling by a low-diversity microbial community in deep groundwater. *Front. Microbiol.* **9**, 1–17 (2018).
46. Ragsdale, S. W. & Pierce, E. Acetogenesis and the Wood-Ljungdahl Pathway of CO₂ Fixation. **1784**, 1873–1898 (2009).
47. Drake, H. *et al.* Extreme fractionation and micro-scale variation of sulphur isotopes during bacterial sulphate reduction in deep groundwater systems. *Geochim. Cosmochim. Acta* **161**, 1–18 (2015).
48. Posiva Oy. *Olkiluoto Site Description 2011*. vol. 31 (2011).
49. Cabello-yeves, P. J. & Rodriguez-valera, F. Marine-freshwater prokaryotic transitions require extensive changes in the predicted proteome. *Microbiome* **7**, (2019).
50. Poirier, S. *et al.* Deciphering intra-species bacterial diversity of meat and seafood spoilage microbiota using *gyrB* amplicon sequencing: A comparative analysis with 16S rDNA V3-V4 amplicon sequencing. *PLoS One* **13**, 1–26 (2018).
51. Caro-Quintero, A. & Ochman, H. Assessing the unseen bacterial diversity in microbial communities. *Genome Biol. Evol.* **7**, 3416–3425 (2015).
52. Heaton, B. E., Herrou, J., Blackwell, A. E., Wysocki, V. H. & Crosson, S. Molecular structure and function of the novel BrnT/BrnA toxin-antitoxin system of *Brucella abortus*. *J. Biol. Chem.* **287**, 12098–12110 (2012).
53. Blanchet, F. G., Cazelles, K. & Gravel, D. Co-occurrence is not evidence of ecological interactions. *Ecol. Lett.* **23**, 1050–1063 (2020).

Reviewers' Comments:

Reviewer #1:

Remarks to the Author:

The authors have addressed most of my concerns. Many clarifications were added, but there are still some issues with the final conclusions. There are several of their conclusions that are still based on speculation. Although the authors acknowledge that they are speculations, these claims are not supported by the data.

In my previous round of comments, I had issues with the term "halt and catch fire". I still think that it is not the correct expression, especially after the authors defined the term, which relates to early operating systems that had to halt and then rebooted to avoid the CPU to fry itself. I think the expression is too strong. Nothing indicates that the cells are too active and they do not need a "reboot". Their data do not provide enough information to even speculate on such a hypothesis.

Writing style:

The authors use the term "multi-omics" (in parentheses) multiple times (lines 57, 66, 80). I think mentioning it once and defining their methods would make it more meaningful and less sounding like a buzz word.

In the new sections, the authors over use the term "interestingly". Such persuading word should not be used as it is used to persuade rather than convey information but rather prompt the reader to thinking it is "interesting".

Line 147: There is no believing in science, only reporting of facts.

Line 167: "somewhat" isn't very descriptive. Change to something ore scientifically meaningful.

Lines 25-26: The gap of knowledge presented in the abstract is that "the evolutionary and ecological constraints on colonization and niche shifts and their consequences for microbiome convergence in oligotrophic groundwaters are unknow". While this is a valid gap of knowledge, this is not a gap that is filled by the study, therefore it is misleading.

Line 31: The authors discuss that there can be a cooperative lifestyle depending on the state of the cells, i.e, aggregate or biofilm. Again, the type of study cannot support such claim. Moreover, it is unclear from the manuscript that they looked at biofilms and aggregates. It is only briefly mentioned that they looked at different fraction sizes.

Lines 124-126: There is mention of different "treatments" and fractions. Usually, "treatments" is used if there was any sort of incubations, which is not the case here. Also, Fig. 1C should highlight these different fraction samples, possibly with some shapes in the MSDS.

Lines 149-151: This new sentence is rather weak. It is very vague. Since the authors do have the data for their samples, they could actually discuss where the possible variations are from.

Lines 173-175: It is unclear why the authors used GyrB to look at diversity rather than ribosomal RNA genes.

Lines 181-183: Not sure what the point of this sentence is. It should be discussed.

Line 216-217: There is no reference for the statement saying that the CRISPR/Cas system is regulated only in the presence of the virus. This is a misleading statement. The CRISPR/Cas system should always be regulated, but in the presence of foreign DNA, the Cas enzymes are cutting DNA.

Lines 221-225: Since the authors do not discuss viral sequences in their data, this statement is very speculative.

Lines 237-238: The authors should define what is meant by "public goods". It is not as obvious as they think it is.

Lines 260-261: Again, not sure what is meant by the burden of public goods.

Lines 315-318: No citation associated with this strong statement.

Line 328: The authors do not propose a model but rather a theory. The authors did not make any statistical correlations between nutrients, geochemical analysis, and expression profiles, therefore their "model" is very speculative.

Lines 340-345: Again, this is not backed up by the data presented, as discussed above.

Figure 3 parts A and B are completely unrelated and should be divided into two figures.

Reviewer #2:

Remarks to the Author:

I thank the authors for taking care of most of my comments during the first round of review. Thus, I only have a few more comments in this round:

L 31-32: The authors again present the biofilm formation and symbiotic interactions as a result, however, this is a conclusion. This needs to be rephrased unless there is real evidence for biofilm formation or symbiotic relationships, e.g. microscopy, co-cultures etc. Also see comment on line 124-126.

L 37-38: The authors still state that there exists a food web in groundwater. This is something that has not been published. In fact, the authors draw these conclusions from a combination of previously published facts, again confounding results and conclusion. Moreover, some of the cited literature doesn't apply: The work by Mike Wilkins and Kelly Wrighton on microbiomes in fracking sites demonstrated that these communities are introduced from the surface and thus their conclusions regarding the food web are not applicable to subsurface ecosystems that were not under anthropogenic influence. This statement needs to be changed, since a conclusion in the beginning of the introduction doesn't make much sense. In addition, the statement is essentially not needed for the story (no need to create room for the study, there is plenty of room already available!).

L124-126: Here the authors say that they have different sample types including biofilm samples and samples from the planktonic community. However, I cannot find anything about these samples in the Material and Methods regarding the sampling strategy for biofilms etc. Please explain and elaborate in the manuscript.

L127-136: Do the organisms belonging to the defined core microbiome share common fast evolving genomic islands (-> prophages, CRISPR arrays etc.)? If so, this would argue against a separation of the sampled ecosystems and explain why the authors find a core microbiome. The finding of a core microbiome is in contrast to previous findings of the deep biosphere, which highlight shared functionality but little taxonomic overlap (Tully et al ISME 2017). Irrespectively, I like the paragraph from line 137-151 discussing the finding of a core microbiome in the deep biosphere.

L249: The authors mention Altiarchaeota and then suggest a heterotrophic and fermentative lifestyle (for the other named taxa, these statements make a lot of sense, of course). I think that Altiarchaeota have been proposed to be autotrophic. How does this relate to the authors' findings?

L334-339: I enjoyed readings this addition a lot!

L342: How does the "Halt and catch fire" concept align with the concept of groundwater, sediment and seed banks that has been proposed by independent labs multiple times (last time I checked

Anantharaman et al 2016 Nature Commun but prior to this also by the groups around Tillmann Luders, Rainer Meckenstock and Christian Griebler)? Please elaborate on this more in the manuscript. Are these concepts synergistic or mutually exclusive?

Supplementary Information L3-4: This statement needs referencing.

Supplementary Information L4-5: This statement needs referencing too.

Supplementary Information L33-48: Something with the referencing seems odd here. Please correct.

Reviewer #3:

Remarks to the Author:

I have now had the opportunity to revisit the manuscript by Mehrshad/Lopez-Fernandez et al. I commend the authors for making extensive revisions. While some of my concerns were addressed, there are several issues that still remain in the manuscript and have not been addressed. Several of these are central to narrative and I describe them below.

1. As I mentioned during my first review, the idea of "Halt and catch fire" is completely unsubstantiated. I see no evidence being added in the revisions to substantiate this claim. While the premise is reasonable – I am not sure as to what this study specifically identifies towards this.

2. I find several flaws in the idea of "public goods". Figure 6 was not provided in the first version, so it was difficult to determine what public goods were being studied. While some of these make sense, I am not sure as to how motility, toxins, and chemotaxis can be considered public goods. Toxins need not be released into the community by cells. Similarly, chemotaxis and motility only impact the cells that are conducting them – I am not sure as to how would this impact the community?

Line 240-244 try to justify the idea of public goods but the statements are completely unsubstantiated. Can authors cite a study to justify the link between constrained space and cheating in the community?

3. Sulfur cycling is central them in the metabolism in the community as presented by the authors. Lines 176-178 mention sulfate reducing bacteria as the predominant metabolic guild driving the flow of carbon.

I do not disagree with the idea. However, the evidence provided to justify this is underwhelming. The authors do not provide any evidence to suggest that the organisms identified as sulfate reducers as in fact sulfate reducers. Figure 5, Supplementary Figure S5 and Supplementary Figure S6 identify some sulfur cycling genes and provide a phylogenetic tree of *dsrA* but no details are provided whatsoever. The tree has no names or accession numbers, the data is not public, and the source of the sequences is not provided. Without this data, it is not possible to evaluate the accuracy of this tree.

I also maintain my criticism from my previous round of review. Given the central role of sulfur cycling that the authors have identified and the presence of MAGs and SAGs for the community, the authors should analyze dissimilatory sulfur metabolism in greater detail. The genes/module studied by the authors is identical between sulfate reducers, sulfur disproportionators, and sulfur oxidizers. While the phylogeny of *dsrA* genes can provide a hint of the direction of metabolism, a better way to analyze this would involve looking at the operon structure and the presence/absence of other genes in the pathway (such as *dsrD*, *dsrE*, *dsrF*, *dsrH*, *dsrS* etc). This is necessary to put the predicted metabolism by the *dsr* pathway in proper context (Sulfate reduction/sulfur oxidation/sulfur disproportionation).

I am not sure as to what Supplementary Table S7 shows. There are seven tabs but I struggled to figure out what the authors are trying to convey.

4. I also encourage the authors to add some descriptions to the supplementary tables. Without any clarification on the different tabs in the tables – they are extremely difficult to interpret.

We appreciate the valuable comments given by the reviewers on our manuscript NCOMMS-20-19610A in this round of revision. We have revised our manuscript according to the reviewer's comments below and provide our point-by-point response to each comment. All changes are highlighted in the submitted track-changes version of the manuscript. Hereby, we quote in italic the reviewers' comments and provide our responses in plain text.

Reviewer #1 (Remarks to the Author):

The authors have addressed most of my concerns. Many clarifications were added, but there are still some issues with the final conclusions. There are several of their conclusions that are still based on speculation. Although the authors acknowledge that they are speculations, these claims are not supported by the data.

C1- *In my previous round of comments, I had issues with the term “halt and catch fire”. I still think that it is not the correct expression, especially after the authors defined the term, which relates to early operating systems that had to halt and then rebooted to avoid the CPU to fry itself. I think the expression is too strong. Nothing indicates that the cells are too active and they do not need a “reboot”. Their data do not provide enough information to even speculate on such a hypothesis.*

In this study, we provide both genomic and transcriptomic evidence for several reversible bacteriostatic functions adapted by the deep biosphere microbiome to cope with their specific niche in the low carbon and energy deep groundwaters. We show that TA system genes in FSGD MAGs/SAGs have greater prevalence compared to MAGs from more shallow aquifers (2402 MAGs from an aquifer adjacent to the Colorado River near Rifle, CO, USA46). The envisioned role of these TA systems in the growth arrest in response to starvation is hypothesized to improve survival during starvation and help with the preservation of common good compounds¹. We also recovered error-prone DNA polymerase (DnaE2) in reconstructed FSGD MAGs/SAGs of 75 genome clusters (223 MAGs/SAGs). These polymerases can be recruited to stalled replication forks and are known to be involved in error-prone DNA damage tolerance² helping with the genome replication and potentially facilitating cell restart after the initial halt enforced by the TA systems. In line with these findings, we hypothesize that deep oligotrophic groundwater microbiota could potentially cope by adapting an episodic lifestyle. Since the reviewer finds the analogy “halt and catch fire” for this episodic lifestyle confusing, we have decided to remove this term from the manuscript and more directly discuss the concept of growth arrest and episodic growth.

Writing style:

C2- *The authors use the term “multi-omics” (in parentheses) multiple times (lines 57, 66, 80). I think mentioning it once and defining their methods would make it more meaningful and less sounding like a buzz word.*

Following reviewer's suggestion, we have now edited this in the text.

C3- *In the new sections, the authors over use the term “interestingly”. Such persuading word should not be used as it is used to persuade rather than convey information but rather prompt the reader to thinking it is “interesting”.*

We have now removed “interestingly” according to reviewer’s suggestion.

C4- *Line 147: There is no believing in science, only reporting of facts.*

We have edited the text and it reads as:

“We recovered 14 MAGs and 1 SAG belonging to this taxon in our FSGD dataset and found it to be one of the highly transcribing taxa in the metatranscriptomes. This further strengthens the concept of a deep biosphere core microbiome that reaches beyond systems with similar lithologies.”

C5- *Line 167: “somewhat” isn’t very descriptive. Change to something ore scientifically meaningful.*

We have now removed “somewhat”. The new text reads as:

“The relative frequency of calculated isoelectric points for predicted proteins in metagenomes sequenced from other drillholes in Olkiluoto and Äspö HRL (ranging from 0.4 to 1.2% salinity) showed a higher signal of basic proteins compared to OL-KR46. The pairwise Kolmogorov’s D values suggests a higher distribution similarity between OL-KR11 and OL-KR13 compared to OL-KR46 as well as a higher distribution similarity between MM, OS, and UM (see **Supplementary Table S6** for pairwise values).”

C6- *Lines 25-26: The gap of knowledge presented in the abstract is that “the evolutionary and ecological constraints on colonization and niche shifts and their consequences for microbiome convergence in oligotrophic groundwaters are unknow”. While this is a valid gap of knowledge, this is not a gap that is filled by the study, therefore it is misleading.*

We have now reworded this sentence for it to specifically reflect the results of our work. The new text reads as:

“While oligotrophic deep groundwaters host active microbes attuned to the low-end of the bioenergetics spectrum, the ecological constraints on microbial niches in these ecosystems and their consequences for microbiome convergence are unknown.”

C7- *Line 31: The authors discuss that there can be a cooperative lifestyle depending on the state of the cells, i.e., aggregate or biofilm. Again, the type of study cannot support such claim. Moreover, it is unclear from the manuscript that they looked at biofilms and aggregates. It is only briefly mentioned that they looked at different fraction sizes.*

We rewrote the abstract section to specify that this hypothesis is based on the transcription profiles where we see transcription of functions related to biofilm formation. The new text reads as:

“Functional expression analysis highlights that these deep groundwater ecosystems foster diverse, yet cooperative communities adapted to this setting. We suggest that they stimulate cooperation by expression of functions related to ecological traits such as aggregate or biofilm formation while alleviating the “tragedy of the commons” by facilitating reciprocal promiscuous metabolic partnerships with other members of the community.”

C8- Lines 124-126: *There is mention of different “treatments” and fractions. Usually, “treatments” is used if there was any sort of incubations, which is not the case here. Also, Fig. 1C should highlight these different fraction samples, possibly with some shapes in the MSDS.*

According to the reviewers’ suggestion we have replaced “treatment: with “sample” in the text and the supplementary information. These different samples are shown in different colors in the Fig. 1C&D and were described in the Supplementary Table S1. For clarity, we have also followed the reviewer’s suggestion by adding a description of these samples in the supplementary information document. The added text reads as:

“We collected groundwater samples from five different boreholes: SA1229A-1 (171.3 mbsl), KA3105A-4 (415.2 mbsl), KA2198A (294.1 mbsl), KA3385A-1 (448.4 mbsl), and KF0069A01 (454.8 mbsl) with varying geochemical conditions as described below. A total of 27 metagenomes were generated from the respective Äspö HRL boreholes that are listed in Supplementary Table S1. The biofilm metagenomes were formed on rock and glass surfaces after 33 days in flow cells attached to the KA2198A and KF0069A01 boreholes (total $n = 8$) as described in Wu et al⁸. Further metagenomes were generated from planktonic cells captured on 0.22 μm filters from boreholes SA1229A, KA3105-4, and KA3385A (total $n = 6$; termed ‘large cells’) and cells that passed through the 0.22 μm filters from same three boreholes ($n = 9$; termed ‘small cells’) as previously described²⁹. Finally, planktonic cells captured on 0.1 μm filters from boreholes SA1229A and KA3385A that were extracted in this study using the MOBIO PowerWater DNA Isolation Kit or phenol-chloroform ($n = 4$).”

C9- Lines 149-151: *This new sentence is rather weak. It is very vague. Since the authors do have the data for their samples, they could actually discuss where the possible variations are from.*

We have now rephrased this sentence and added references to make it clearer. The new version reads as:

“However, presence of this core microbiome in different deep groundwater ecosystems would vary due to physicochemical³ and geological factors that affect the composition of the local biomes.”

C10- Lines 173-175: *It is unclear why the authors used GyrB to look at diversity rather than ribosomal RNA genes.*

This analysis was specifically requested by the reviewer #2 in the first round of revision. Please see the comment: “L 148: *Species richness cannot be determined from binned genomes. Maybe that one sample with low richness was problematic in good binning? Please add an analysis of species richness of all samples based on marker genes. Do not use ribosomal RNA for that as it doesn’t get assembled well. Use a house keeping gene that is good for species delineation, e.g. gyrB*”. We agreed with the reviewer to also use GyrB for representing the diversity as it could potentially assemble better from metagenomes compared to 16S rRNA.

C11- Lines 181-183: *Not sure what the point of this sentence is. It should be discussed.*

We have now rephrased this sentence and added discussion to make it clearer. It now reads as:

“Additionally, genome Cluster 25 includes seven representatives of *Pseudodesulfovibrio aespoensis* (originally classified as *Desulfovibrio aespoensis*)⁴ that was originally isolated from 600 mbsl in Äspö HRL groundwater. Genome cluster 25 affiliated to this species contains MAGS from the Olkiluoto drillholes OL-KR13 and OL-KR46 extending the presence of representatives of this species to both studied sites.”

C12- Line 216-217: *There is no reference for the statement saying that the CRISPR/Cas system is regulated only in the presence of the virus. This is a misleading statement. The CRISPR/Cas system should always be regulated, but in the presence of foreign DNA, the Cas enzymes are cutting DNA.*

We agree with the reviewer and have now reworded this sentence to make it clear. It now reads:

“Naturally, these processes are active in the case of exposure to viruses or other mobile genetic elements.”

C13- Lines 221-225: *Since the authors do not discuss viral sequences in their data, this statement is very speculative.*

We have removed this statement as per reviewer’s suggestion. It now reads as:

“The expression profile of this SAG serves as an example of the importance of selective transcription in this deep groundwater microbe to fine-tune its response to selfish genetic elements.”

C14- Lines 237-238: *The authors should define what is meant by “public goods”. It is not as obvious as they think it is.*

As per reviewers’ suggestion we have now added a concise description of public goods and added related references to this section. The added text reads as:

“Microbial public goods are loosely defined as functions or products that are costly to produce for an individual and provide collective benefit for the surrounding community. While these common goods can have different forms, they are generally released into the extracellular environment⁵.”

C15- Lines 260-261: *Again, not sure what is meant by the burden of public goods.*

Here we refer to the burden imposed on the public good producers due to exploitation of their products by ‘cheaters’. The latter are groups that profit from public goods without contributing to their production and can thus be an energetic and metabolic burden for the public good producers. Several strategies are hypothesized to be used by the public good producers to ensure cooperation in the presence of such cheaters. These include privatization, spatial structure, group selection, horizontal gene transfer, etc.

In deep groundwaters, taxa such as Patescibacteria and DPANN archaea depend on an array of such public good compounds produced by other groups. They ensure cooperation by engaging in reciprocal partnerships such as supplying fermentation products (e.g., lactate, acetate, and hydrogen), vitamins, amino acids, and secondary metabolites to direct or indirect partners in their immediate surroundings. We have now reworded this section to make the message clearer. It now reads:

“Hence, to stabilize their own symbiotic niches, these cells likely participate in reciprocal partnerships where they supply fermentation products (e.g., lactate, acetate, and hydrogen), vitamins, amino acids, and secondary metabolites to their direct or indirect partners in their immediate surroundings.”

C16- Lines 315-318: *No citation associated with this strong statement.*

The idea of spore formation being as a ‘dead-end strategy’ has been advocated by Bo Barker Jørgensen and the text has been edited to add the reference and to clarify that the further example of a negative aspect of spore formation is discussion. The text now reads:

“Sporulation might be a potential mechanism for these MAGs to cope with environmental stressors such as low carbon and energy conditions. However, this survival mechanism has been suggested to be a ‘dead-end strategy’ in the long-term due to e.g. the need for cell repair upon revival⁶. An additional possible detrimental aspect may include the response time to a transient nutrient and energy supply that may be consumed by active cells before the spore has germinated.”

C17- Line 328: The authors do not propose a model but rather a theory. The authors did not make any statistical correlations between nutrients, geochemical analysis, and expression profiles, therefore their “model” is very speculative.

This is more of a conceptual model based on multiple lines of evidence from genomic data as explained in the manuscript. We thank the reviewer for their suggestion and have now used “theory” instead of model.

C18- Lines 340-345: Again, this is not backed up by the data presented, as discussed above.

We present both genomics and transcriptomics evidence for several reversible bacteriostatic functions adapted by the deep biosphere microbiome to cope with their specific niche in the low carbon and energy deep groundwaters. We show that TA system genes in FSGD MAGs/SAGs have greater prevalence compared to MAGs from more shallow aquifers (2402 MAGs from an aquifer adjacent to the Colorado River near Rifle, CO, USA⁴⁶). The envisioned role of these TA systems in the growth arrest in response to starvation is hypothesized to improve survival during starvation and help with the preservation of common good compounds¹. We also recovered error-prone DNA polymerase (DnaE2) in reconstructed FSGD MAGs/SAGs of 75 genome clusters (223 MAGs/SAGs). These polymerases can be recruited to stalled replication forks and are known to be involved in error-prone DNA damage tolerance² helping with the genome replication and potentially facilitating cell restart after the initial halt enforced by the TA systems. In line with these findings, we hypothesize that deep oligotrophic groundwater microbiota could potentially cope by adapting an episodic lifestyle. As the reviewer #1 finds the analogy “halt and catch fire” for this episodic lifestyle confusing, we have decided to remove this term from the manuscript and more directly discuss the concept of growth arrest and episodic growth.

We have now moved this paragraph to the conclusion section of the manuscript. As we mention in the manuscript this conclusion is based on genomic and metatranscriptomic information there is a need for more empirical work to confirm or refute these predictions. We hope that this will satisfy reviewer’s concern.

C19- Figure 3 parts A and B are completely unrelated and should be divided into two figures. We respectfully disagree with the reviewer in this specific instance. In this figure we wish to highlight the presence/absence pattern of different genome clusters in different metagenomes and at the same time show how these genome clusters are transcribed in the metatranscriptomes generated from the different borehole samples. We think presenting these two panels together in one figure have added value for the readers.

Reviewer #2 (Remarks to the Author):

I thank the authors for taking care of most of my comments during the first round of review. Thus, I only have a few more comments in this round:

C20- L 31-32: *The authors again present the biofilm formation and symbiotic interactions as a result, however, this is a conclusion. This needs to be rephrased unless there is real evidence for biofilm formation or symbiotic relationships, e.g. microscopy, co-cultures etc. Also see comment on line 124-126.*

We have now rephrased this section to address reviewer's concern. In our results we see transcription of functions related to biofilm formation in our transcriptomes (shown in Fig. 6) and in this section we are referring to these results. According to the genomic information we see many metabolic deficiencies in the representatives of Patescibacteriota and DPANN Archaea as none of them encode genomic potential for a complete fatty acid (FA)-based lipid biosynthesis pathway yet still these cells have fatty acids in their membrane. This has been discussed in a study by Probst & Elling et al.⁷ where they also mention "Other putative bacterial and archaeal symbionts from different branches of the tree of life also do not encode for their own lipid biosynthesis pathway and at least one hyperthermophilic episymbiont (*Nanoarchaeum equitans*) has been suggested to acquire its lipids from the host archaeon. However, the origin and types of lipids used by CPR bacteria remain elusive." And additionally we find "representatives of four highly expressing Patescibacteriota genome clusters (Clusters 41, 436, 587, and 596) contain genes encoding for phosphate acetyltransferase and acetate kinase functions. These proteins facilitate production of acetate from acetyl CoA via a two-step pathway where acetyl phosphate occurs as an intermediate. A fifth highly expressing Patescibacteriota (cluster 594) may also feature this pathway but only the acetate kinase gene was detected (annotations provided in **Supplementary Table S10**). The production of acetate by Patescibacteriota genome clusters could potentially supply Desulfobacterota and UBA9089 representatives with a primary carbon and electron source and form the basis for a reciprocal partnership." While these evidences highlight a symbiotic relationship, since we do not have culture or microscopic proof for this we have now rephrased this section. The updated text reads as:

"Functional expression analysis highlights that these deep groundwater ecosystems foster diverse, yet cooperative communities adapted to this setting. We suggest that they stimulate cooperation by expression of functions related to ecological traits such as aggregate or biofilm formation while alleviating the "tragedy of the commons" by facilitating reciprocal promiscuous metabolic partnerships with other members of the community."

C21- L 37-38: *The authors still state that there exists a food web in groundwater. This is something that has not been published. In fact, the authors draw these conclusions from a combination of previously published facts, again confounding results and conclusion. Moreover, some of the cited literature doesn't apply: The work by Mike Wilkins and Kelly Wrighton on microbiomes in fracking sites demonstrated that these communities are introduced from the surface and thus their conclusions regarding the food we are not applicable to subsurface ecosystems that were not under anthropogenic influence. This statement needs to be changed, since a conclusion in the beginning of the introduction doesn't make much sense. In addition, the statement is essentially not needed for the story (no need to create room for the study, there is plenty of room already available!).*

We have now removed the "food web" from this sentence following reviewer's suggestion. The new sentence reads as:

"Archaea and Bacteria are a critical component of deep groundwater ecosystems that host all domains of life as well as viruses⁸⁻¹¹."

C22- L124-126: *Here the authors say that they have different sample types including biofilm samples and samples from the planktonic community. However, I cannot find anything about these samples in the*

Material and Methods regarding the sampling strategy for biofilms etc. Please explain and elaborate in the manuscript.

Information about sample types were mentioned in the Supplementary Table S1. To make this information more accessible we have now summarized this in the Supplementary Information document. The added text reads as:

“We collected groundwater samples from five different boreholes: SA1229A-1 (171.3 mbsl), KA3105A-4 (415.2 mbsl), KA2198A (294.1 mbsl), KA3385A-1 (448.4 mbsl), and KF0069A01 (454.8 mbsl) with varying geochemical conditions as described below. A total of 27 metagenomes were generated from the respective Äspö HRL boreholes that are listed in Supplementary Table S1. The biofilm metagenomes were formed on rock and glass surfaces after 33 days in flow cells attached to the KA2198A and KF0069A01 boreholes (total $n = 8$) as described in Wu et al¹². Further metagenomes were generated from planktonic cells captured on 0.22 μm filters from boreholes SA1229A, KA3105-4, and KA3385A (total $n = 6$; termed ‘large cells’) and cells that passed through the 0.22 μm filters from same three boreholes ($n = 9$; termed ‘small cells’) as previously described¹³. Finally, planktonic cells captured on 0.1 μm filters from boreholes SA1229A and KA3385A that were extracted in this study using the MOBIO PowerWater DNA Isolation Kit or phenol-chloroform ($n = 4$).”

C23- L127-136: Do the organisms belonging to the defined core microbiome share common fast evolving genomic islands (-> prophages, CRISPR arrays etc.)? If so, this would argue against a separation of the sampled ecosystems and explain why the authors find a core microbiome. The finding of a core microbiome is in contrast to previous findings of the deep biosphere, which highlight shared functionality but little taxonomic overlap (Tully et al ISME 2017). Irrespectively, I like the paragraph from line 137-151 discussing the finding of a core microbiome in the deep biosphere.

We would like to thank the reviewer for the positive feedback. The samples compared are geographically (hundreds of kilometers) well separated where any connectivity between e.g. deep Fennoscandian groundwaters in Finland and southern Sweden is highly unlikely.

As for the comparison with the Tully et al publication¹⁴ (Tully, B. J., Wheat, C. G., Glazer, B. T. & Huber, J. A. A dynamic microbial community with high functional redundancy inhabits the cold, oxic subseafloor aquifer. ISME J. 12, 1–16 (2018)) they mainly focus on the community composition dynamics for several sampling points from CORK fluid delivery lines that are 5950m apart. While the title and the main discussion focuses on the dynamic community and functional redundancy, in the results section “Assessment of microorganisms in the subseafloor aquifer” the authors mention “Based on the relative abundance of sequencing reads competitively recruited to each genome from each sample, the mean relative abundance for all genomes in all samples was 0.19% (median, 0.004%), and when examined closely, most genomes were ‘present’ in all samples at low abundance values (<0.05%; on average 151 of 195 genomes were below this threshold in each sample). NORP9 had the highest relative abundance (40.4%) in the 2012 U1383C deep sample (Figure 3; Supplementary Data 9). Genomes were subjected to Bray-Curtis clustering based on the relative abundance values (Supplementary Figure 3). Several of the genomes (for example, NORP125, -161, and -172) were cosmopolitan in the subseafloor crustal fluids, present in both holes, and at several time points and depths”.

And they continue by mentioning “Generally, when genomes were grouped together, these groups were abundant in one or a few samples (e.g., NORP51, -54 and -55). When groups of organisms are present in

multiple samples, the samples tend to be in close spatial proximity or sequential sampling events (Figure 3).”

Tully et al focus on the microbial community composition dynamics and address the functional redundancy. Nevertheless, they briefly touch upon the recovery of genomes that are present in both these subseafloor locations. We do not think that findings of these two studies are in contrast. These two studies deal with quite different ecosystems and the degree of overlap in the common core microbiome could be different. The two subterrestrial deep groundwater locations of our study are geographically well separated (hundreds of kilometers) and finding a core common microbiome in such disconnected ecosystems is a novel finding.

C24- L249: *The authors mention Altiarchaeota and then suggest a heterotrophic and fermentative lifestyle (for the other named taxa, these statements make a lot of sense, of course). I think that Altarchaeota have been proposed to be autotrophic. How does this relate to the authors’ findings?*

We double checked the MAGs/SAGs affiliated to this phylum and none of them encode the complete archaeal type of the reductive acetyl-CoA (Wood-Ljungdahl) pathway. This could be due to MAG/SAG incompleteness or this metabolism might not be present in all representatives of this phylum. According to the GTDB taxonomy, our MAGs/SAGs of this phylum are members of new taxa at the family or genus level and they can potentially have different metabolisms than the other known autotrophic Altarchaeota. To make sure that we reflect this valid concern of the reviewer in the text we have now reworded this section and it reads as:

“However, exploitation of public goods seems inevitable in groundwaters considering the high phylogenetic diversity and widespread presence of e.g. Patescibacteriota representatives (300 MAGs and 19 SAGs forming 152 clusters) as well as DPANN representatives¹⁵ including Nanoarchaeota representatives (100 MAGs in 56 clusters in classes Aenigmarchaeia, Nanohaloarchaea, CG03, and Woesearchaeia), and Micrarchaeota (15 MAGs in 9 clusters). The metabolic context of these reconstructed MAGs/SAGs suggests a primarily heterotrophic and fermentative lifestyle lacking core biosynthetic pathways for nucleotides, amino acids, and lipids. This is in agreement with prior reports on Patescibacteriota and DPANN representatives¹⁶ that are highly dependent on their adjacent cells to supply them with metabolites they cannot synthesize themselves^{7,16,17}. Members of Altiarchaeota among the DPANN have been shown to be autotrophs capable of fixing carbon dioxide by using a modified version of the reductive acetyl-CoA (Wood-Ljungdahl) pathway¹⁸. However, FSGD MAGs and SAGs affiliated to the Altiarchaeota phylum (6 MAGs and 2 SAGs in 2 clusters) are suggested to be heterotrophic as they lack the full Wood-Ljungdahl pathway. This could be due to MAG/SAG incompleteness or that this metabolism is not consistently present in all representatives of this phylum.”

C25- L334-339: *I enjoyed readings this addition a lot!*

We thank the reviewer for the positive feedback.

C26- L342: *How does the “Halt and catch fire” concept align with the concept of groundwater, sediment and seed banks that has been proposed by independent labs multiple times (last time I checked Anantharaman et al 2016 Nature Commun but prior to this also by the groups around Tillmann Luders, Rainer Meckenstock and Christian Griebler)? Please elaborate on this more in the manuscript. Are these concepts synergistic or mutually exclusive?*

The conceptual model of an episodic lifestyle suggested for the deep oligotrophic groundwater here on the basis of our large-scale genomic observations could be an expansion of previous ideas of a seed bank. In most cases while discussing a seed bank, dormancy is considered equal to spore formation¹⁹ and other dormancy mechanisms and their distribution among different cells are not discussed in detail. In our study, we show that other reversible bacteriostatic strategies (such as TA systems explained in this study) could play a notable role in the oligotrophic deep groundwaters and should be empirically explored. We have inspected our reconstructed MAGs and SAGs and only a handful of them contain the genes necessary for spore formation. This is detailed on line 329 and reads “Sporulation related genes are present in the MAGs KR46_Ju.mb.14, KR46_M_MetaG.mb.3, KR46_S_MetaG.mb.7, and MM_PW_MetaG.mb.36 out of the 17 MAGs/SAGs representatives of the phyla Firmicutes, Firmicutes_A, and Firmicutes_B (**Supplementary Table S10**)”.

Our analysis of the metatranscriptomes show “that a small yet significant portion of FSGD MAG/SAG clusters are actively transcribing in the nine sequenced metatranscriptomes derived from the three Äspö HRL boreholes (ca. 26% of genome clusters in MM-171.3, 14% in TM-448.4, and 5% in MM-415.2 metatranscriptomes; **Fig. 3**)”. This raises the question about how successful the spore forming taxa would be in catching ephemeral pulses of nutrient supplies while there exist active members of the community competing for the same resource? This reiterates the concerns of others who have also considered the possibility that the spore formation in extremely low energy and nutrient ecosystems could be a ‘dead-end strategy’⁶ where microbes are on a slow march to death¹⁹.

We here provide an explanation for how some other deep groundwater taxa incapable of spore formation cope with low nutrient and energy conditions. By providing genomic and transcriptomic proof for the reversible bacteriostatic approaches such as TA systems, we expand the understanding of the seed bank concept in the deep oligotrophic groundwaters by suggesting an episodic lifestyle for microbes. We have added a summary of this explanations to the conclusion section of the manuscript. The added text reads as:

“By providing genomic and transcriptomic proof for the dormancy via reversible bacteriostatic functions such as TA systems, we expand the understanding of the ecological aspect of the microbial seed bank concept²⁰ in the deep oligotrophic groundwaters by suggesting an active but episodic strategy for the phylogenetically diverse microbiome of the deep groundwater in response to ephemeral nutrient pulses.”

C27- Supplementary Information L3-4: This statement needs referencing.

References were added to the text.

C28- Supplementary Information L4-5: This statement needs referencing too.

References were added to the text.

The mentioned paragraph with added reference:

During the last decades, it has been demonstrated that microorganisms inhabit the sub-surface biome down to several kilometers below the surface²¹. Although microbial life in the deep crust biosphere has been gaining interest²², due to its difficulty to access it is still one of the least understood environments on earth²³. Consequently, many novel taxa are being identified in the deep biosphere^{24,25}.

C29- Supplementary Information L33-48: *Something with the referencing seems odd here. Please correct. Thank you for pointing this out we have now fixed this reference formatting issue.*

Reviewer #3 (Remarks to the Author):

I have now had the opportunity to revisit the manuscript by Mehrshad/Lopez-Fernandez et al. I commend the authors for making extensive revisions. While some of my concerns were addressed, there are several issues that still remain in the manuscript and have not been addressed. Several of these are central to narrative and I describe them below.

C30- 1. *As I mentioned during my first review, the idea of “Halt and catch fire” is completely unsubstantiated. I see no evidence being added in the revisions to substantiate this claim. While the premise is reasonable – I am not sure as to what this study specifically identifies towards this.*

In this study, we provide both genomics and transcriptomics evidence for several reversible bacteriostatic functions adapted by the deep biosphere microbiome to cope with their specific niche in the low carbon and energy deep groundwaters. We show that TA system genes in FSGD MAGs/SAGs have greater prevalence compared to MAGs from more shallow aquifers (2402 MAGs from an aquifer adjacent to the Colorado River near Rifle, CO, USA46). The envisioned role of these TA systems in the growth arrest in response to starvation is hypothesized to improve survival during starvation and help with the preservation of common good compounds¹. We also recovered error-prone DNA polymerase (DnaE2) in reconstructed FSGD MAGs/SAGs of 75 genome clusters (223 MAGs/SAGs). These polymerases can be recruited to stalled replication forks and are known to be involved in error-prone DNA damage tolerance² helping with the genome replication and potentially facilitating cell restart after the initial halt enforced by the TA systems. While all these findings presented in the manuscript support our hypothesis that deep oligotrophic groundwater microbiota could potentially cope by adapting an episodic lifestyle, we agree with the reviewer that this is not conclusive evidence and because of this we have consistently been careful not to make any such claims. Since also reviewer #1 found the analogy “halt and catch fire” confusing, we have decided to remove it from the manuscript and more directly discuss the concept of growth arrest and episodic growth.

C31- 2. *I find several flaws in the idea of “public goods”. Figure 6 was not provided in the first version, so it was difficult to determine what public goods were being studied. While some of these make sense, I am not sure as to how motility, toxins, and chemotaxis can be considered public goods. Toxins need not be released into the community by cells. Similarly, chemotaxis and motility only impact the cells that are conducting them – I am not sure as to how would this impact the community?*

As we are mainly focusing on the transcribed functions related to biofilm, we have now updated this figure and removed these three categories from the figure. We have also updated the Supplementary Table S8.

Figure 6- Expression level of some functional classes involved in public good provision in the sequenced metatranscriptomes. The list of screened KO identifiers is shown in **Supplementary Table S8**.

However, just as a clarification functions such as toxins have been categorized as public good. Of course this does not apply to all toxins types but those secreting from the cell and those involved in pathogenicity has been suggested as public good (please see Table 1 in the Smith & Schuster primer publication⁵). Chemotaxis and motility can be a part of the complex network of signaling pathways that can respond to nutrient concentrations and consequently affect the community-wide response to public goods and cooperation vs competition at large (Box 1 in the Wadhams & Armitage review for some examples)²⁶.

C32- Line 240-244 try to justify the idea of public goods but the statements are completely unsubstantiated. Can authors cite a study to justify the link between constrained space and cheating in the community?

In here the “spatially constrained” refers to the structure of biofilm where adjacent cooperating cells are exposed to higher concentrations of the public goods released from a cell and then have a better chance of assimilating them. This can allow kin selection to favor public good production and cooperation²⁷. As an example, for references in this context we pick “Drescher, K., Nadell, C. D., Stone, H. A., Wingreen, N. S. & Bassler, B. L. Solutions to the public goods dilemma in bacterial biofilms. *Curr. Biol.* 24, 50–55 (2014)”

We have now rephrased this section to make it clearer. The new text reads as:

“One way to alleviate the “tragedy of the commons” imposed by the production and sharing of such public goods is by emergence of local sub-communities through biofilm or aggregate formation²⁷. Functions related to biofilm formation are detected in the transcription profiles (0.5 to 2% of the total transcription profile) of the deep groundwaters studied here. Biofilm formation could potentially help with reducing the number of cheaters and provide an evolutionary advantage for cooperation as compared to competition²⁷.”

C33- 3. Sulfur cycling is central them in the metabolism in the community as presented by the authors. Lines 176-178 mention sulfate reducing bacteria as the predominant metabolic guild driving the flow of carbon.

*I do not disagree with the idea. However, the evidence provided to justify this is underwhelming. The authors do not provide any evidence to suggest that the organisms identified as sulfate reducers are in fact sulfate reducers. Figure 5, Supplementary Figure S5 and Supplementary Figure S6 identify some sulfur cycling genes and provide a phylogenetic tree of *dsrA* but no details are provided whatsoever. The tree has no names or accession numbers, the data is not public, and the source of the sequences is not provided. Without this data, it is not possible to evaluate the accuracy of this tree.*

We have uploaded the *dsrA* alignment and phylogeny tree to figshare and they are publicly available at <https://doi.org/10.6084/m9.figshare.14166638> and <https://doi.org/10.6084/m9.figshare.14166650.v1> respectively. These links have now been inserted in the figure legend for the Supplementary Figure S5. We have also added a PDF of the *dsrA* tree to the “source data” folder in the submission for the reviewer to inspect the reference sequences used to reconstruct this phylogeny.

*I also maintain my criticism from my previous round of review. Given the central role of sulfur cycling that the authors have identified and the presence of MAGs and SAGs for the community, the authors should analyze dissimilatory sulfur metabolism in greater detail. The genes/module studied by the authors is identical between sulfate reducers, sulfur disproportionators, and sulfur oxidizers. While the phylogeny of *dsrA* genes can provide a hint of the direction of metabolism, a better way to analyze this would involve looking at the operon structure and the presence/absence of other genes in the pathway (such as *dsrD*, *dsrE*, *dsrF*, *dsrH*, *dsrS* etc). This is necessary to put the predicted metabolism by the *dsr* pathway in proper context (Sulfate reduction/sulfur oxidation/sulfur disproportionation).*

Following reviewer’s suggestion, we have now expanded on this section by analyzing all MAGs that contain a *dsrA* gene for the presence of *sat*, *aprAB*, *dsrC*, *dsrEFH*, and *dsrD* genes. Detailed methods of this analysis have been added to the Supplementary information document and results are now mentioned in the text and included as **Supplementary Table S9**. We have now elaborated more on the dissimilatory sulfur metabolism based on the results of this additional analysis to the highest resolution possible by inspecting the genomic features of MAGs and SAGs. However, differentiating these metabolisms in MAGs and SAGs has its own limitation as it is mentioned by Anantharaman et. al. “Sulfur disproportionating organisms cannot be differentiated from sulfite-reducing organisms on genomic features alone”²⁸. In those cases, we denoted both possibilities in the presented results. Upon adding this new more thorough analysis we have now removed the **Supplementary Figure S6** to avoid unnecessary repetition.

Added text reads as:

Methods in the Supplementary information document:

Dissimilatory sulfur metabolism. The phylogeny of *DsrA* as a key gene in the dissimilatory sulfur metabolism was generated by using reference *dsrA* genes (both oxidative and reductive types) together with the genes annotated as *dsrA* in the reconstructed MAGs/SAGs of our study (> 200 amino acids). Sequences were aligned using Muscle²⁹ in MEGA7³⁰ and evolutionary relationships were visualized by constructing a maximum-likelihood phylogenetic tree (JTT +CAT model). All residues were used, and the tree was bootstrapped with 100 replicates. MAGs and SAGs containing *dsrA* gene were further inspected for the presence of *aprAB* (K00394 and K00395), *sat* (K00958), *dsrAB* (K11180 and K11181), *dsrC* (K11179), *dsrD* (PF08679), and *dsrEFH* (K07235, K07236, and K07237). These genes were searched for using eggno-

mapper³¹ (v. 2.2.1 with the eggno5.0 database) and pfam_scan.pl (v. 1.6 with the 31.0 release of the PFAM database) and annotations were manually validated for each gene. The contribution of each MAG/SAG to the sulfur cycle was inferred according to the pattern of presence/absence of these genes as suggested by Anantharaman et. al²⁸. For the final inference members of each cluster were considered together.

The results in the main text read as:

“Reconstructing the metabolic scheme of highly expressing genome clusters, apart from small heterotrophic cells with a proposed symbiotic lifestyle (**Fig. 5**), highlights a central role of sulfur as an electron acceptor that is commonly used in the energy metabolism of the deep groundwater microbiome³². A total of 189 MAGs/SAGs contain the dissimilatory sulfite reductase A/B subunits (*dsrAB*), of which 48 branch together with the oxidative and 141 cluster with the reductive *dsrA* reference proteins in the reconstructed phylogeny (**Supplementary Fig. S5**). We further inspected these MAGs and SAGs for genes related to sulfur metabolism (sulfate adenylyltransferase (*sat*), adenylylsulfate reductase subunits A/B (*aprAB*), dissimilatory sulfite reductase A/B subunits (*dsrAB*), *dsrC*, *dsrD*, and *dsrEFH*) to determine the direction of dissimilatory sulfur metabolism in these organisms³³. Oxidative *dsrA* genes belonged to 19 genome clusters affiliated to the Proteobacteria phylum. Among these, 17 clusters contribute to the sulfur cycle via sulfur oxidation to sulfate and the genome cluster 85 represents the genomic capacity for sulfur oxidation to sulfite. Genome Cluster 329 is missing *dsrEFH* genes and consequently the direction of its sulfur metabolism could not be confidently assigned from the genomic content of reconstructed SAG (**Supplementary Table S9**).

Among MAGs and SAGs containing the reductive type of *dsrA*, 115 MAGs/SAGs affiliated to phyla Chloroflexota, Desulfobacterota, Firmicutes-B, and Nitrospirota contribute to the sulfur cycle via sulfate reduction to sulfide (**Supplementary Table S9**) whereas for five Desulfobacterota representatives the genomic information could not differentiate between sulfite reduction to sulfide and sulfur disproportionation. A total of 21 MAGs/SAGs (affiliated to phyla AABM5-125-24, Actinobacteriota, Desulfobacterota, Nitrospirota, SAR324, UBA9089, UBP1, and Zixibacteria) contain the genes *aprAB*, *sat*, and the reductive type of *dsrA* but are missing *dsrD*. While the absence of *dsrD* gene in these MAG/SAG could be due to incompleteness, representatives of the listed taxa have previously been reported to be capable of sulfite/sulfate reduction³³ and feature DsrA proteins that branch together with the reductive DsrA reference proteins in our reconstructed phylogeny (**Supplementary Fig. S5 and Supplementary Table S9**). Representatives of 19 clusters (ca. 22%) of the highly expressing groups represent genomic capacity for dissimilatory sulfur metabolism (**Fig. 5**).”

C34- I am not sure as to what Supplementary Table S7 shows. There are seven tabs but I struggled to figure out what the authors are trying to convey.

The supplementary Table S7 shows the gene annotation for the transcribed CDSs in MAGs undefined_mixed_planktonic.mb.19 and MM_PC_MetaG.mb.230 along with SAG 3300021839. We represent detailed annotations based on NCBI conserved domains and HMMER. We explored the variable codon utilization among highly expressing representatives of MAGs/SAGs affiliated with phylum Desulfobacterota by separately calculating codon frequency for expressing CDSs. These MAGs and SAG have different synonymous codon frequency and distribution among the expressed portion of the genome and the rest of the genome. We further explored the expressed genes to identify the functions being expressed.

We have now reshaped the Supplementary Table S7 to make it more accessible. We also added a description box in the first tab in the table.

C35- 4. I also encourage the authors to add some descriptions to the supplementary tables. Without any clarification on the different tabs in the tables – they are extremely difficult to interpret.

Following reviewer’s suggestion, we have now added a description box in the first sheet of supplementary tables containing multiple tabs to explain their content.

References

1. Magnuson, R. D. Hypothetical functions of toxin-antitoxin systems. *J. Bacteriol.* **189**, 6089–6092 (2007).
2. Alves, I. R. *et al.* Effect of SOS-induced levels of imuABC on spontaneous and damage-induced mutagenesis in *Caulobacter crescentus*. *DNA Repair (Amst)*. **59**, 20–26 (2017).
3. He, C. *et al.* Genome-resolved metagenomics reveals site-specific diversity of episymbiotic CPR bacteria and DPANN archaea in groundwater ecosystems. *Nat. Microbiol.* (2021) doi:10.1038/s41564-020-00840-5.
4. Motamedi, M. & Pedersen, K. *Desulfovibrio aespoensis* sp. nov., a mesophilic sulfate-reducing bacterium from deep groundwater at Aspo hard rock laboratory, Sweden. *Int. J. Syst. Bacteriol.* **48**, 311–315 (1998).
5. Smith, P. & Schuster, M. Public goods and cheating in microbes. *Curr. Biol.* **29**, R442–R447 (2019).
6. Jørgensen, B. B. Shrinking majority of the deep biosphere. *Proc. Natl. Acad. Sci. U. S. A.* **109**, 15976–15977 (2012).
7. Probst, A. J. *et al.* Lipid analysis of CO₂-rich subsurface aquifers suggests an autotrophy-based deep biosphere with lysolipids enriched in CPR bacteria. *ISME J* 1547–1560 (2020) doi:10.1038/s41396-020-0624-4.
8. Lopez-Fernandez, M. *et al.* Metatranscriptomes reveal that all three domains of life are active but are dominated by Bacteria in the Fennoscandian crystalline granitic continental deep biosphere. *MBio* **9**, 1–15 (2018).
9. Daly, R. *et al.* Viruses control dominant bacteria colonizing the terrestrial deep biosphere after hydraulic fracturing. *Nat. Microbiol.* **4**, 352–361 (2019).
10. Kyle, J. E., Eydal, H. S. C., Ferris, F. G. & Pedersen, K. Viruses in granitic groundwater from 69 to 450 m depth of the spö hard rock laboratory, Sweden. *ISME J.* **2**, 571–574 (2008).
11. Eydal, H. S. C., Jägevall, S., Hermansson, M. & Pedersen, K. Bacteriophage lytic to *Desulfovibrio aespoensis* isolated from deep groundwater. *ISME J.* **3**, 1139–1147 (2009).
12. Wu, X. *et al.* Potential for hydrogen-oxidizing chemolithoautotrophic and diazotrophic populations to initiate biofilm formation in oligotrophic, deep terrestrial subsurface waters. *Microbiome* **5**, 1–13 (2017).
13. Wu, X. *et al.* Microbial metagenomes from three aquifers in the Fennoscandian shield terrestrial deep biosphere reveal metabolic partitioning among populations. *ISME J.* **10**, 1192–1203 (2016).
14. Tully, B. J., Wheat, C. G., Glazer, B. T. & Huber, J. A. A dynamic microbial community with high functional redundancy inhabits the cold, oxic subseafloor aquifer. *ISME J.* **12**, 1–16 (2018).
15. Dombrowski, N., Lee, J., Williams, T. A., Offre, P. & Spang, A. Genomic diversity, lifestyles and evolutionary origins of DPANN archaea. *FEMS Microbiol. Lett.* **366**, fnz008 (2019).
16. Castelle, C. J. *et al.* Biosynthetic capacity, metabolic variety and unusual biology in the CPR and DPANN radiations. *Nat. Rev. Microbiol.* **16**, 629–645 (2018).
17. Castelle, C. J. & Banfield, J. F. Major new microbial groups expand diversity and alter our understanding of the tree of life. *Cell* **172**, 1181–1197 (2018).

18. Probst, A. J. *et al.* Biology of a widespread uncultivated archaeon that contributes to carbon fixation in the subsurface. *Nat. Commun.* **5**, (2014).
19. Shoemaker, W. R. & Lennon, J. T. Evolution with a seed bank: The population genetic consequences of microbial dormancy. *Evol. Appl.* **11**, 60–75 (2018).
20. Lennon, J. T. & Jones, S. E. Microbial seed banks: The ecological and evolutionary implications of dormancy. *Nat. Rev. Microbiol.* **9**, 119–130 (2011).
21. Magnabosco, C. *et al.* The biomass and biodiversity of the continental subsurface. *Nat. Geosci.* **11**, 707–717 (2018).
22. Colman, D. R., Poudel, S., Stamps, B. W., Boyd, E. S. & Spear, J. R. The deep, hot biosphere: Twenty-five years of retrospection. *Proc. Natl. Acad. Sci.* **114**, 6895–6903 (2017).
23. Cario, A., Oliver, G. C. & Rogers, K. L. Exploring the Deep Marine Biosphere: Challenges, Innovations, and Opportunities. *Front. Earth Sci.* **7**, 1–9 (2019).
24. Sackett, J. D. *et al.* Four draft Single-Cell genome sequences of novel , nearly identical Kiritimatiellaeta strains isolated from the continental deep subsurface. *Microbiol Resour Announc* e01249-18 (2019).
25. Anantharaman, K. *et al.* Thousands of microbial genomes shed light on interconnected biogeochemical processes in an aquifer system. *Nat. Commun.* **7**, 1–11 (2016).
26. Wadhams, G. H. & Armitage, J. P. Making sense of it all: Bacterial chemotaxis. *Nat. Rev. Mol. Cell Biol.* **5**, 1024–1037 (2004).
27. Drescher, K., Nadell, C. D., Stone, H. A., Wingreen, N. S. & Bassler, B. L. Solutions to the public goods dilemma in bacterial biofilms. *Curr. Biol.* **24**, 50–55 (2014).
28. Anantharaman, K. *et al.* Expanded diversity of microbial groups that shape the dissimilatory sulfur cycle. *ISME J.* **12**, 1715–1728 (2018).
29. Edgar, R. C. MUSCLE: multiple sequence alignment with high accuracy and high throughput. *Nucleic Acid Res.* **32**, 1792–1797 (2004).
30. Kumar, S., Stecher, G. & Tamura, K. MEGA7: Molecular Evolutionary Genetics Analysis Version 7.0 for Bigger Datasets. *Mol. Biol. Evol.* **33**, 1870–1874 (2016).
31. Huerta-Cepas, J. *et al.* Fast genome-wide functional annotation through orthology assignment by eggNOG-mapper. *Mol. Biol. Evol.* **34**, 2115–2122 (2017).
32. Bell, E. *et al.* Active sulfur cycling in the terrestrial deep subsurface. *ISME J.* **14**, 1260–1272 (2020).
33. Anantharaman, K. *et al.* Expanded diversity of microbial groups that shape the dissimilatory sulfur cycle. *ISME J.* **12**, 1715–1728 (2018).

Reviewers' Comments:

Reviewer #1:

Remarks to the Author:

I appreciate that the authors have addressed all of my comments and the publication is much improved.

I only have one more comment:

Line 245: Not sure if bacteria can be "cheaters" as this personifies the microbes.

Reviewer #2:

Remarks to the Author:

The authors have answered all my queries to my satisfaction. I recommend publication.

Reviewer #3:

Remarks to the Author:

I commend the authors for taking into account my comments and conducting detailed revisions. Amongst my comments, specifically the authors have (1) toned down the terminology and unsubstantiated claims, and (2) analyzed sulfur metabolism in greater detail which backs up the previously made claims made in the manuscript. Additional figures and methods have also been added in my line with the changes. I do not have any more concerns.

I only have one outstanding comment. I was not able to access the NCBI Bioproject mentioned in the manuscript, please ensure that the data is released in a timely manner in NCBI. Also, to increase the utility of their data, I would encourage the authors to add a table linking Genome name to NCBI Genome accession number (may be in Supp table 1 ?). While the figures contain names, I did not see a link to accession numbers anywhere.

Dear Editor and reviewers;

We have revised our manuscript to incorporate the reviewer's comments and provide our point-by-point response to each comment. All changes are highlighted in the submitted track-changes version of the manuscript.

Reviewer #1:

I appreciate that the authors have addressed all of my comments and the publication is much improved.

We would like to thank the reviewer for their positive assessment of our work.

I only have one more comment:

Line 245: Not sure if bacteria can be "cheaters" as this personifies the microbes.

We have now modified this to "microbes that profit from common goods without contributing to these shared resources".

Reviewer #2:

The authors have answered all my queries to my satisfaction. I recommend publication.

We would like to thank the reviewer for their comments and suggestions during the review process.

Reviewer #3:

I commend the authors for taking into account my comments and conducting detailed revisions. Amongst my comments, specifically the authors have (1) toned down the terminology and unsubstantiated claims, and (2) analyzed sulfur metabolism in greater detail which backs up the previously made claims made in the manuscript. Additional figures and methods have also been added in my line with the changes. I do not have any more concerns.

We thank the reviewer for their thorough revision of our manuscript.

I only have one outstanding comment. I was not able to access the NCBI Bioproject mentioned in the manuscript, please ensure that the data is released in a timely manner in NCBI. Also, to increase the utility of their data, I would encourage the authors to add a table linking Genome name to NCBI Genome accession number (may be in Supp table 1 ?). While the figures contain names, I did not see a link to accession numbers anywhere.

All MAGs and SAGs of this study are publicly available for batch download via figshare with the DOI: <https://doi.org/10.6084/m9.figshare.12170313.v1>. This has been addressed in the Data Availability section of the manuscript.

We have also deposited the same MAGs to the GenBank and have contacted the NCBI staff to make it publicly available. They have been successfully released to GenBank and will all be available under the bioproject PRJNA627556: Fennoscandian Shield genomic database (FSGD) with the same sample name as the MAG identifiers used in this manuscript that makes it more user-friendly. We added detailed information of how to access the data to the Data Availability section of the manuscript. And following reviewer's suggestion we have now added a tab to the Supplementary Data 1 including the GenBank accession number for each MAG.

However, working with large data ourselves we believe the most user-friendly way for batch download of the whole dataset together with alignments and phylogenies supporting our findings is via figshare. This dataset has been publicly available when our preprint referring to this DOI: <https://doi.org/10.6084/m9.figshare.12170313.v1> became online at the beginning of the review process and has been downloaded several times already.